# CUDA-L1: Improving CUDA Optimization via Contrastive Reinforcement Learning

**Xiaoya Li, Xiaofei Sun, Albert Wang, Jiwei Li and Chris Shum**

**DeepReinforce Team**

 github.com/deepreinforce-ai/CUDA-L1

## Abstract

The exponential growth in demand for GPU computing resources has created an urgent need for automated CUDA optimization strategies. While recent advances in LLMs show promise for code generation, current state-of-the-art models achieve low success rates in improving CUDA speed. In this paper, we introduce CUDA-L1, an automated reinforcement learning (RL) framework for CUDA optimization that employs a novel contrastive RL algorithm.

CUDA-L1 achieves significant performance improvements on the CUDA optimization task: trained on NVIDIA A100, it delivers an average speedup of $\times$**3.12** with a median speedup of $\times$**1.42** against default baselines over across all 250 CUDA kernels of KernelBench, with peak speedups reaching $\times$**120**. In addition to the default baseline provided by KernelBench, CUDA-L1 demonstrates $\times$**2.77** over Torch Compile, $\times$**2.88** over Torch Compile with reduce overhead, and $\times$**2.81** over CUDA Graph implementations. Furthermore, the model also demonstrates portability across GPU architectures. CUDA-L1 opens possibilities for automated optimization of CUDA operations, and holds promise to substantially promote GPU efficiency and alleviate the rising pressure on GPU computing resources. 

## 1 Introduction

The exponential growth in demand for GPU computing resources, driven primarily by the rapid advancement and deployment of Large Language Models (LLMs), has created an urgent need for highly efficient CUDA optimization strategies. Traditionally, CUDA optimization has been a highly manual and time-intensive process, where skilled engineers must meticulously analyze memory access patterns, experiment with different thread block configurations, and iteratively profile their code through extensive trial-and-error cycles.

Recent advances in LLMs (Team et al., 2023; Shengyu et al., 2023; Team et al., 2024; Grattafiori et al., 2024; Yang et al., 2025; Hurst et al., 2024; Jiang et al., 2024; Liu et al., 2024a; OLMo et al., 2024), especially those powered with RL (Jaech et al., 2024; Guo et al., 2025; Wang et al., 2024; Muennighoff et al., 2025), have demonstrated remarkable capabilities in code generation and algorithm design. Despite the promise, current performance remains limited. State-of-the-art LLM models such as DeepSeek-R1 (Guo et al., 2025) and OpenAI-o1 (Jaech et al., 2024) achieve low success rates in generating optimized CUDA code (only approximately 15% on KernelBench (Ouyang et al., 2025)) To address these limitations and unlock the potential of LLMs for automated CUDA optimization, in this work, we propose CUDA-L1, an LLM framework powered by contrastive reinforcement learning for CUDA optimization. CUDA-L1 is a pipelined framework, the core of which is a newly-designed contrastive RL framework.

Different from previous RL models (Williams, 1992; Shao et al., 2024; Schulman et al., 2017) , contrastive RL performs comparative analysis of previously generated CUDA variants alongside their execution performance, enabling the model to improving through distinguishing between effective and ineffective optimization strategies. Contrastive-RL simultaneously optimizes the foundation

---

 Email: {xiaoya_li, xiaofei_sun, albert_wang, jiwei_li, chris_shum}@deep-reinforce.com

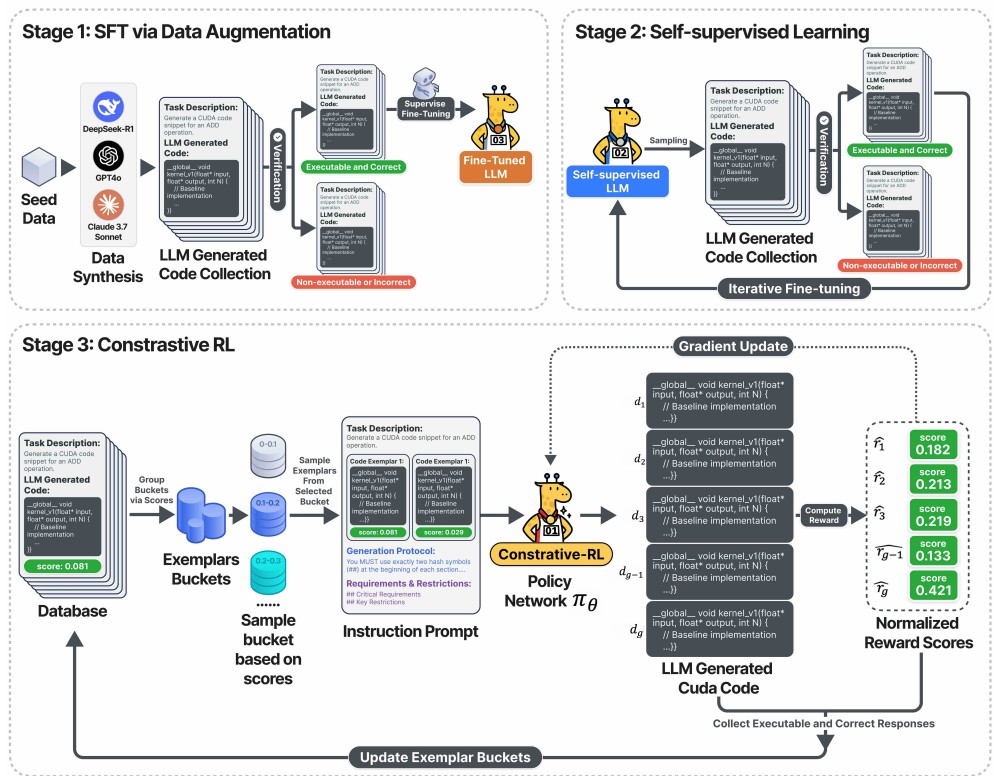

Figure 1: Overview of the CUDA-L1 training pipeline.

model through gradient-based parameter updates while fulfilling the maximum potential from the current model through contrastive analysis from high-performance CUDA variants, creating a co-evolutionary dynamic that drives superior CUDA optimization performance.

CUDA-L1 delivers significant improvements on the CUDA optimization task: trained on NVIDIA A100, it achieves an **average** speedup of ×**3.12** (**median** ×**1.42**) over the default baseline across all 250 KernelBench CUDA kernels, with maximum speedups reaching ×**120**. In addition, CUDA-L1 demonstrates ×**2.77** over Torch Compile, ×**2.88** over Torch Compile with reduce overhead, ×**2.81** over CUDA Graph implementations. Furthermore, the CUDA codes optimized specifically for A100 demonstrate strong portability across GPU architectures, with similar optimization patterns observed across different baseline configurations: achieving average speedups of ×**3.85** (median ×**1.32**) on H100, ×**3.13** (median ×**1.31**) on L40, ×**2.51** (median ×**1.18**) on RTX 3090, and ×**2.38** (median ×**1.34**) on H20. Similar performance improvements over Torch Compile, Torch Compile with reduce overhead, CUDA Graph are consistently observed across all GPU types.

CUDA-L1 reveals a remarkable capability of RL in autonomous learning for CUDA optimization:

1. Even starting with a foundation model with poor CUDA optimization ability, by using code speedups as RL rewards and proper contrastive RL training techniques, we can still train an RL system capable of generating CUDA optimization codes with significant speedups.
2. Without human prior knowledge, RL systems can independently discover CUDA optimization techniques, learn to combine them strategically, and more importantly, extend the acquired CUDA reasoning abilities to unseen kernels. This capability unlocks the potential for a variety of automatic CUDA optimization tasks, e.g., kernel parameter tuning, memory access pattern optimization, and different hardware adaptations, offering substantial promises to enhance GPU utilization.

Another contribution of this work is the enrichment of the KernelBench dataset with CUDA Graph implementations. Please refer to supplementary materials. We release these implementations to the community, providing substantially stronger baselines for performance comparison.

## 2 CUDA-L1

### 2.1 OVERVIEW

Existing large language models (Guo et al., 2025; Yang et al., 2025; Grattafiori et al., 2024) demonstrate significant limitations in generating executable and correct CUDA code with speedup, as reported in prior research (Ouyang et al., 2025). This deficiency likely stems from the insufficient representation of CUDA code in the training datasets of these models. To address this fundamental gap, we introduce a three-stage pipelined training strategy for CUDA-L1, i.e., Supervised fine-tuning via data augmentation, Self-supervised learning and Contrastive reinforcement learning, aiming to progressively enhances the model's CUDA programming capabilities:

Before we delve into the details of each stage, we provide key definitions adopted throughout the rest of this paper:

1. **Executability**: A CUDA code is executable if it successfully compiles, launches, and executes to completion within $1000\times$ the runtime of the reference implementation. Code exceeding this runtime threshold is considered unexecutable.[1]
2. **Correctness**: A CUDA code is correct if it produces equivalent outputs to the reference implementation across 1000 random test inputs.[2]
3. **Success**: A CUDA code is successful if it is both executable and correct.

### 2.2 SFT VIA DATA AUGMENTATION

In the SFT stage, we collect a dataset by using existing LLMs to generate CUDA code snippets and selecting successful one. This dataset is directly used to fine-tune the model. Throughout this paper, we use deepseek-v3-671B (Liu et al., 2024a) as the model backbone. Please refer to the details of data collection in Appendix A.1. The collected dataset $D$ is used to finetune the foundation model. The instruction to the model is the same as the prompt for dataset generation, where the reference code $q_i$ is included in the instruction and the model is asked to generate an improved version. The model is trained to predict each token in $d_{i,j}$ given the instruction.

### 2.3 SELF-SUPERVISED LEARNING

Now we are presented with the finetuned model after the SFT stage, where the model can potentially generate better CUDA code with higher success rates than the original model without finetuning. We wish to further improve the model's ability to generate successful CUDA code by exposing it to more code snippets generated by itself.

We achieve this iteratively by sampling CUDA code from the model, evaluating it for executability and correctness, removing the unsuccessful trials and keeping the successful ones. Successful ones are batched and used to update the model parameters. Using the updated model, we repeat the process: generating code, evaluating it, and retraining the model. It is worth noting that during the self-supervised learning stage, we focus exclusively on the executability and correctness of the generated code, without considering speed as a metric. This design choice reflects our primary objective of establishing reliable code generation before optimizing for performance.

### 2.4 CONTRASTIVE REINFORCEMENT LEARNING

Now we have a model capable of generating successful CUDA code at a reasonable success rate. Next, we aim to optimize for execution speed.

One straightforward approach is to apply existing reinforcement learning algorithms such as RE-INFORCE (Williams, 1992), GRPO (Shao et al., 2024), or PPO (Schulman et al., 2017). In this approach, we would ask the model to first perform chain-of-thought reasoning (Wei et al., 2022), then generate code, evaluate it, and use the evaluation score to update the model parameters. However, our experiments reveal that these methods perform poorly in this task. The issue is as follows: standard RL algorithms compute a scalar reward for each generated CUDA code sample. During training, this reward undergoes algorithm-specific processing (e.g., baseline subtraction in REINFORCE, advantage normalization in GRPO, importance sampling in PPO). The processed reward then serves as a loss weighting term for gradient updates, increasing the likelihood of high-reward

---

[1]This threshold is reasonable since code with $1000\times$ slower performance contradicts our speedup optimization goals.

[2]Prior work uses only 5 random inputs, which we found insufficient for robust validation.

sequences while decreasing the likelihood of low-reward sequences. Critically, in this paradigm, the reward signal is used exclusively for parameter updates and is never provided as input to the LLM. Consequently, the LLM cannot directly reason about performance trade-offs during code generation. To address this limitation, we propose incorporating reward information directly into the reasoning process by embedding performance feedback within the input prompt. Specifically, we present the model with multiple code variants alongside their corresponding speedup scores. Rather than simply generating code, the LLM is trained to first conduct comparative analysis of why certain implementations achieve superior performance, then synthesize improved solutions based on these insights. Each generated code sample undergoes evaluation to obtain a performance score, which serves dual purposes in our training framework: the primary purpose is to the score acts as a reward signal for gradient-based parameter optimization, updating model weights. The score functions as a reward signal for gradient-based parameter optimization, directly updating the model weights; the other is to construct prompt for future training stages. The scored code sample becomes part of the exemplar set for subsequent training iterations, enriching the contrastive learning dataset.

This dual-utilization strategy enables iterative optimization across two complementary dimensions:

**Foundation Model Enhancement**   Parameter updates progressively improve the model's fundamental understanding and capabilities for CUDA optimization tasks, expanding its representational capacity.

**Fixed-Parameter Solution Optimization**   The contrastive approach seeks to extract the maximum potential from the current model's parameters by leveraging comparative analysis of high-quality exemplars.

These two optimization processes operate synergistically: enhanced foundation models enable more accurate contrastive reasoning, while improved reasoning strategies provide higher-quality training signals for parameter updates of foundation models. This co-evolutionary dynamic drives convergence toward optimal performance. We term this approach contrastive reinforcement learning (contrastive-RL for short).

### 2.4.1   PROMPT CONSTRUCTION

Here we describe the construction of prompts provided to the LLM. The prompt provided to the LLM during Contrastive-RL training comprises the following structured components:

 I) **Task Descrption:** A detailed description of the computational problem, including input/output specifications, performance requirements, and optimization objectives.
 II) **Previous Cuda Codes with Scores:** Previously generated CUDA implementations paired with their corresponding performance scores (e.g., execution time, throughput, memory efficiency), providing concrete examples of varying solution quality.
 III) **Generation Protocol:** Explicit instructions defining the required output format and components.
 IV) **Requirements and Restrictions**: Requirements and restrictions to prevent reward hacking in RL.

The model's response must contain the following three structured components:

 I) **Performance Analysis:** A comparative analysis identifying which previous kernel implementations achieved superior performance scores and the underlying algorithmic or implementation factors responsible for success.
 II) **Algorithm Design:** A high-level description of the proposed optimization strategy, outlining the key techniques to be applied, presented as numbered points in natural language.
 III) **Code Implementation:** The complete CUDA kernel implementation incorporating optimizations.

A detailed demonstration for the prompt in shown in Table 9.

### 2.4.2   CONTRASTIVE EXEMPLAR SELECTION

The selection of code exemplars for prompt construction is critical, as core of Contrastive-RL is to perform meaningful comparative analysis. The selection strategy needs to addresses the following two key requirements: first, for achieving competitive performance, the exemplar set should include higher-performing implementations to guide the model toward competitive CUDA codes, avoiding local minima that result from analyzing and comparing inferior codes; second, to ensure

performance diversity, the selected codes must exhibit substantial performance differences to enable effective contrastive analysis.

We employ a sampling strategy akin to that adopted by evolutionary LLM models: Let $N$ denote the number of code exemplars included in each prompt (set to $N = 2$ in our experiments). During RL training, we maintain a performance-indexed database of all successful code samples generated during RL training. Codes are organized into performance buckets $B_k$ based on discretized score intervals, where bucket $B_i$ contains codes with scores in range $[s_k, s_k + \Delta s)$.

We first sample $N$ distinct buckets according to a temperature-scaled softmax distribution:

$$P(B_i) = \frac{\exp\left((\bar{s}_i - \mu_s)/\tau\right)}{\sum_j \exp\left((\bar{s}_j - \mu_s)/\tau\right)} \tag{1}$$

where $\bar{s}_i$ denotes the aggregate score of bucket $B_i$, computed as the mean of its constituent code scores, $\mu_s = \text{mean}(\{\bar{s}_j\}_{j=1}^M)$ represents the global mean of all bucket scores, and $\tau$ is the temperature parameter governing the exploration-exploitation tradeoff. The sampling strategy in Equation 1 differs from conventional temperature sampling in evolutionary LLM approaches through a modification: the deduction of $\mu_s$ stabilizes the distribution by centering scores around zero, which prevents absolute score magnitudes from dominating the selection.

From each selected bucket $B_i$, we uniformly sample one representative code to construct the final prompt set. This approach satisfies both design criteria: Regarding competitive Performance, score-weighted bucket sampling biases selection toward higher-performing implementations, ensuring the exemplar set contains competitive solutions; Regarding performance Diversity, enforcing selection from $N$ distinct buckets ensures sufficient performance variance for effective contrastive analysis.

A more sophisticated alternative is to use an island-based approach for exemplar selection. However, we find no significant difference in performance between our bucket-based method and the island-based approach. Given this, we opt for the simpler bucket-based strategy.

### 2.4.3 REWARD

In this subsection, we detail the computation of the execution time-based reward function, which serves dual purposes: (1) guiding parameter updates in reinforcement learning and (2) constructing effective prompts. Given a reference CUDA implementation $q_i$ from PyTorch with successful execution time $t_{q_i}$, and a generated code candidate $d$ with execution time $t_d$, we define the single-run speedup score as:

$$\text{r}_{\text{single-run}}(d) = \frac{t_{q_i}}{t_d} \tag{2}$$

More details for denoising the rewards are shown in Appendix A.6.

For RL training, we adopt the Group Relative Policy Optimization (GRPO) strategy (Shao et al., 2024). Please refer to Appendix A.3 for details.

## 3 EXPERIMENTS AND ANALYSIS

### 3.1 KERNELBENCH AND EVALUATION

Our evaluation is conducted on the KernelBench dataset (Ouyang et al., 2025). The KernelBench Dataset contains a collection of 250 PyTorch workloads designed to evaluate language models' ability to generate efficient GPU kernels. The dataset is structured across three hierarchical levels based on computational complexity: Level 1 contains 100 tasks with single primitive operations (such as convolutions, matrix multiplications, activations, and normalizations), Level 2 includes 100 tasks with operator sequences that can benefit from fusion optimizations (combining multiple operations like convolution + ReLU + bias), and Level 3 comprises 50 full ML architectures sourced from popular repositories including PyTorch, Hugging Face Transformers, and PyTorch Image Models (featuring models like AlexNet and MiniGPT). Each task in the dataset provides a reference PyTorch implementation with standardized input/output specifications, enabling automated evaluation of both functional correctness and performance through wall-clock timing comparisons. The dataset represents real-world engineering challenges where successful kernel optimization directly translates to practical performance improvements. Throughout this paper, we use KernelBench as the evaluation benchmark. KernelBench is recognized as a challenging benchmark in the community (Ouyang et al., 2025), with even the best current LLMs improving fewer than 20% of tasks.

| Configuration | Method | Mean | Max | 75% | 50% | 25% | Success↑ # out of total | Speedup↑ >1.01x out of total |
|---|---|---|---|---|---|---|---|---|
| *Default* | All | 3.12 | 120 | 2.25 | 1.42 | 1.17 | 249/250 | 226/250 |
| | Level 1 | 2.78 | 65.8 | 1.75 | 1.28 | 1.12 | 99/100 | 80/100 |
| | Level 2 | 3.55 | 120 | 2.05 | 1.39 | 1.20 | 100/100 | 98/100 |
| | Level 3 | 2.96 | 24.9 | 2.60 | 1.94 | 1.42 | 50/50 | 48/50 |
| *Torch Compile* | All | 2.77 | 69.0 | 2.55 | 1.72 | 1.14 | 249/250 | 203/250 |
| | Level 1 | 3.04 | 59.7 | 2.71 | 1.99 | 1.41 | 99/100 | 89/100 |
| | Level 2 | 2.91 | 69.0 | 1.99 | 1.55 | 1.10 | 100/100 | 78/100 |
| | Level 3 | 1.98 | 8.57 | 2.28 | 1.68 | 1.00 | 50/50 | 36/50 |
| *Torch Compile RO* | All | 2.88 | 80.1 | 2.48 | 1.67 | 1.13 | 249/250 | 200/250 |
| | Level 1 | 3.38 | 55.3 | 3.02 | 2.29 | 1.61 | 99/100 | 90/100 |
| | Level 2 | 3.00 | 80.1 | 2.06 | 1.54 | 1.10 | 100/100 | 79/100 |
| | Level 3 | 1.62 | 8.67 | 1.76 | 1.13 | 0.991 | 50/50 | 31/50 |
| *CUDA Graph* | All | 2.81 | 97.9 | 1.83 | 1.20 | 0.954 | 249/250 | 147/229 |
| | Level 1 | 3.18 | 59.6 | 2.09 | 1.38 | 1.04 | 99/100 | 68/88 |
| | Level 2 | 2.84 | 97.9 | 1.55 | 1.08 | 0.932 | 100/100 | 53/94 |
| | Level 3 | 2.06 | 24.6 | 1.74 | 1.08 | 0.887 | 50/50 | 26/47 |

Table 1: Performance comparison across different configurations on KernelBench on A100. RO = Reduce Overhead. Success and Speedup indicate the number of successful benchmarks out of the total for each level. Note that for CUDA Graph, the total benchmark count differs from the dataset/data-subset size, as some original reference code in KernelBench cannot be successfully transformed into the corresponding CUDA Graph implementations.

For each task with reference implementation $q$, we evaluate the performance of a generated CUDA code $d$ using a similar protocol to training: We execute both $q$ and $d$ in randomized order within a fixed time budget of 20 minutes per task. The number of execution rounds varies across tasks due to differences in individual runtimes. The final evaluation score for $d$ is computed as the average speedup ratio across all execution rounds within the allocated time window. Unsuccessful implementations receive a score of zero. The metrics we report include speedup statistics (mean, maximum, and 75th, 50th, and 25th percentiles), success rate, and percentage of improvements.

## 3.2 COMPARISON SETUPS

To perform a comprehensive evaluation on the generated code, we perform the following comparisons:

**I) Default**    This compares the CUDA-L1 generated code with the reference code by KernelBench.

**II) Torch Compile**    This compares the CUDA-L1 generated code with the reference code enhanced by torch.compile with default settings. Torch.compile applies graph-level optimizations including operator fusion, memory planning, and kernel selection to accelerate PyTorch models through just-in-time compilation.

**III) Torch Compile Reduce Overhead**    This compares the CUDA-L1 generated code with the reference code enhanced by torch.compile with reduce-overhead mode enabled. This mode minimizes the compilation overhead by caching compiled graphs more aggressively and reducing recompilation frequency, making it particularly suitable for inference workloads with static shapes.

**IV) CUDA Graph**    Since KernelBench does not provide official CUDA Graph implementations, we employ Claude 4 to generate CUDA Graph-augmented code for each reference implementation. CUDA Graphs capture a series of CUDA kernels and their dependencies into a single graph structure that can be launched with minimal CPU overhead, eliminating the need for repeated kernel launch commands and significantly reducing CPU-GPU synchronization costs. Specifically, we provide Claude 4 with the reference code and request the addition of CUDA Graph optimizations. The generated output is then evaluated for correctness. If the code fails validation, we iterate by providing Claude 4 with both the original reference code and the previous erroneous outputs, requesting a corrected version. This iterative process continues for up to 10 attempts until the generated code passes all correctness checks. We release the CUDA Graph codes for KernelBench to the community, pro-

| Methods | Model | Mean | Max | 75% | 50% | 25% | Success↑ # out of 250 | Speedup↑ >1.01 # out of 250 |
|---|---|---|---|---|---|---|---|---|
| *Vanilla* | Llama 3.1-405B | 0.23 | 3.14 | 0.63 | 0 | 0 | 68 | 5 |
| | DeepSeek-V3 | 0.34 | 2.96 | 0.76 | 0 | 0 | 99 | 9 |
| | DeepSeek-R1 | 0.88 | 14.4 | 1.00 | 0.75 | 0 | 179 | 18 |
| | OpenAI-O1 | 0.73 | 12.4 | 1.00 | 0.55 | 0 | 141 | 14 |
| *Evolve* | Llama 3.1-405B | 1.18 | 18.4 | 1.03 | 1.00 | 1.00 | 247 | 88 |
| | DeepSeek-V3 | 1.32 | 52.4 | 1.32 | 1.03 | 1.00 | 247 | 113 |
| | DeepSeek-R1 | 1.41 | 44.2 | 1.45 | 1.17 | 1.00 | 248 | 162 |
| | OpenAI-O1 | 1.35 | 63.9 | 1.38 | 1.16 | 1.00 | 247 | 158 |
| *CUDA-L1* | Stage 1 | 1.14 | 32.7 | 1.00 | 1.00 | 0.96 | 240 | 50 |
| | Stage 1+2 | 1.36 | 48.3 | 1.41 | 1.09 | 1.00 | 247 | 175 |
| | Stage 1+2+GRPO | 2.41 | 84.6 | 1.83 | 1.33 | 1.11 | 247 | 207 |
| | 3 stages - random | 2.14 | 64.5 | 1.62 | 1.21 | 1.09 | 241 | 186 |
| | - island | **3.21** | **126** | 2.21 | 1.40 | 1.16 | **249** | 223 |
| | - bucket | 3.12 | 120 | **2.25** | **1.42** | **1.17** | **249** | **226** |

Table 2: Model performances on KernelBench All Level.

viding researchers and practitioners with ready-to-use optimized implementations that can serve as strong baselines for future performance studies and benchmarking efforts.

## 3.3 MAIN RESULTS ON KERNELBENCH

The experimental results in Table 1 demonstrate CUDA-L1's optimization effectiveness across different baseline configurations on KernelBench. CUDA-L1 achieves substantial performance improvements over the Default baseline with $3.12\times$ average speedup and $120\times$ maximum gains. Against Torch compilation baselines, CUDA-L1 delivers moderate but consistent improvements with $2.77$–$2.88\times$ mean speedup ratios, while demonstrating $2.81\times$ mean improvement over CUDA Graph baseline with notable $97.9\times$ maximum gains.

Across difficulty levels, CUDA-L1's optimization effectiveness varies by task complexity. For Level 1 (single operations), CUDA-L1 achieves moderate improvements ranging from $2.78$–$3.38\times$ over different baselines. Level 2 (operator sequences) shows CUDA-L1's strongest performance with $3.55\times$ improvement over Default baseline. Level 3 (complex ML tasks) reveals interesting baseline-dependent effectiveness: CUDA-L1 achieves $2.96\times$ improvement over Default baseline, but shows reduced effectiveness against Torch compilation baselines (only $1.62$–$1.98\times$ improvements), suggesting these configurations provide stronger baseline performance for complex operations.

## 3.4 BASELINE COMPARISON

We compare the results with the following three groups of baselines:

**Vanilla Foundation Models:** To establish baseline performance benchmarks, we evaluate OpenAI-o1, DeepSeek-R1, DeepSeek-V3, and Llama 3.1-405B Instruct (denoted by **OpenAI-o1-vanilla**, **DeepSeek-R1-vanilla**, **DeepSeek-V3-vanilla** and **Llama 3.1-405B-vanilla**) by prompting each model to optimize the reference CUDA code. The generated CUDA code is directly used for evaluation without further modification. For each task, we repeat this process 5 times and report the best.

**Evolutionary LLM**: We implement evolutionary LLM strategies where, given a set of previous codes, we sample up to 4 high-performing kernels based on evaluation scores. The key difference is that the model only performs contrastive analysis without updating model parameters. We adopt the island strategy for code database construction and sampling, as suggested in Novikov et al. (2025). We conduct experiments on DeepSeek-R1, OpenAI-o1 and and Llama 3.1-405B, denoted as **DeepSeek-R1-evolve**, **OpenAI-o1-evolve**, **DeepSeek-V3-evolve** and **Llama 3.1-405B-evolve**.

**Different combinations of CUDA-L1 components and variants:**

- **stage1**: Uses only the outcome from the first stage with supervised fine-tuning applied
- **stage1+2**: Applies only the first two stages without reinforcement learning
- **stage1+2 + GRPO**: Replaces the contrastive RL with a vanilla GRPO strategy, without comparative analysis

| Configuration | GPU Device | Mean | Max | 75% | 50% | 25% | Success ↑ | Speedup ↑ |
|---|---|---|---|---|---|---|---|---|
| | | | | | | | # out of 250 | >1.01x |
| *Default* | A100 | 3.12 | 120 | **2.25** | **1.42** | **1.17** | 249 | **226**/250 |
| | 3090 | 2.51 | 114 | 1.57 | 1.18 | 1.03 | 242 | 201/250 |
| | H100 | **3.85** | **368** | 1.76 | 1.32 | 1.09 | **250** | 218/250 |
| | H20 | 2.38 | 63.7 | 1.81 | 1.34 | 1.11 | 247 | **226**/250 |
| | L40 | 3.13 | 182 | 1.88 | 1.31 | 1.08 | 248 | 215/250 |
| *Torch Compile* | A100 | 2.77 | 69.0 | 2.55 | 1.72 | 1.14 | 249 | 203/250 |
| | 3090 | 2.58 | 73.2 | 2.23 | 1.50 | 1.00 | 242 | 177/250 |
| | H100 | 2.74 | 49.7 | 2.83 | 1.92 | 1.11 | **250** | 195/250 |
| | H20 | **2.89** | 49.4 | **3.21** | **2.04** | **1.19** | 247 | **209**/250 |
| | L40 | 2.85 | **96.9** | 2.43 | 1.82 | 1.13 | 248 | 199/250 |
| *Torch Compile RO* | A100 | 2.88 | 80.1 | 2.48 | 1.67 | **1.13** | **249** | **200**/250 |
| | 3090 | 2.61 | 72.9 | 2.29 | 1.48 | 1.00 | 242 | 172/250 |
| | H100 | 2.77 | 61.2 | 2.78 | 1.61 | 1.00 | 247 | 187/250 |
| | H20 | 2.82 | 52.1 | **3.18** | 1.64 | 1.06 | 247 | 192/250 |
| | L40 | **2.89** | **90.9** | 2.54 | **1.72** | 1.08 | 248 | 193/250 |
| *CUDA Graph* | A100 | 2.81 | 97.9 | 1.83 | 1.20 | 0.954 | **229** | 147/229 |
| | 3090 | 3.34 | 156 | **1.94** | **1.28** | **0.997** | 206 | **148/206** |
| | H100 | 2.23 | 70.1 | 1.60 | 1.04 | 0.838 | 222 | 119/222 |
| | H20 | 2.20 | 64.6 | 1.69 | 1.09 | 0.854 | **229** | 133/229 |
| | L40 | **3.98** | **275** | 1.83 | 1.16 | 0.862 | 224 | 137/224 |

Table 3: Performance comparison across different configurations and GPU devices on KernelBench. RO = Reduce Overhead. Speedup is defined as value exceeding 1.01x.

- **random sampling**: Replaces the bucket sampling strategy with simple random sampling of exemplars
- **island sampling**: Adopts an island-based sampling strategy Novikov et al. (2025), where examples are distributed across different islands, prompts are constructed using exemplars from the same island, and newly generated examples are added to that island. After a fixed number of iterations, examples in half of the inferior islands are eliminated and examples from superior islands are copied to replace them.

Results are shown in Table 2. As observed, all vanilla foundation models perform poorly on this task. Even the top-performing models, such as DeepSeek-R1 and OpenAI-o1, achieve speedups over the reference kernels in fewer than 10% of tasks, while Llama 3.1-405B optimizes only 2.4% of tasks. This confirms that vanilla foundation models cannot be readily applied to CUDA optimization due to their insufficient grasp of CUDA programming principles and optimization techniques.

We observe significant performance improvements introduced by the Evolutionary LLM models compared to vanilla foundation model setups, despite sharing the same parameter sets. All Evolve models achieve speedups in over 70% of tasks, with DeepSeek-R1 reaching 72.4% success rate. This demonstrates that leveraging contrastive analysis, which exploits the model's general reasoning abilities, is more effective than direct output generation. The superiority of evolutionary LLM over vanilla LLM also provides evidence that contrastive RL should outperform non-contrastive RL approaches like vanilla GRPO, as the relationship between evolutionary and vanilla LLMs parallels that between contrastive and non-contrastive RL methods.

When comparing the different combinations of CUDA-L1 components, we observe a progressive increase in speedup rates from **stage1 (SFT only)** at 22.4% to **stage1+2 (SFT + self-supervised)** at 66%, and further to **stage1+2+GRPO** at 88.4%. This demonstrates the cumulative benefits of each training stage in improving model performance.

All RL-based approaches significantly outperform evolutionary LLM baselines with fixed model parameters, with the best RL methods achieving over 95% speedup rates compared to 72.4% for the best evolutionary approach. This demonstrates the necessity of model parameter updating for achieving optimal performance in CUDA optimization tasks.

### 3.5 GENERALIZATION OF A100-OPTIMIZED KERNELS TO OTHER GPU ARCHITECTURES

Even without being specifically tailored to other GPU architectures, we observe significant performance improvements across all tested GPU types, with mean speedups ranging from $2.38\times$ to $3.85\times$. H100 achieves the highest mean speedup ($3.85\times$) with exceptional maximum gains ($368\times$), while A100 PCIe and L40 demonstrate strong performance with mean speedups of $3.12\times$ and $3.13\times$ respectively. L40 shows the second-highest maximum speedup ($182\times$) among all GPUs. The consumer RTX 3090 achieves a competitive mean speedup of $2.51\times$, while H20 shows moderate performance with $2.38\times$ mean speedup. Notably, A100 maintains the highest 75th percentile ($2.25\times$), 50th percentile ($1.42\times$), and 25th percentile ($1.17\times$) values, indicating more consistent optimization performance on the target architecture.

The success rates remain high across all architectures (242-250 out of 250), with H100 achieving perfect success (250/250), validating that CUDA optimization techniques can generalize across different GPU architectures. Speedup achievement rates ($>1.01\times$) vary by architecture, with H20 and A100 showing the highest effectiveness (226 and 226 successful optimizations respectively), while RTX 3090 demonstrates good performance with 201 successful optimizations.

These results demonstrate that while A100-optimized kernels transfer to other GPUs with varying degrees of effectiveness, the optimizations achieve substantial improvements across architectures. H100's exceptional performance suggests strong compatibility with the optimization techniques, while A100's consistent percentile performance validates the target architecture optimization. The varying maximum speedups ($63.7\times$ to $368\times$) across GPUs indicate architecture-specific optimization potential, suggesting that dedicated optimizations for each GPU type would further enhance performance. We plan to release kernels specifically trained for different GPU types in an updated version of CUDA-L1.

## 4 RELATED WORK

### 4.1 RL-AUGMENTED LLMS FOR CODE OPTIMIZATION

Starting this year, there has been a growing interest in using LLM or RL-augmented LLM models for code optimization, including recent work on compiler optimization (Cummins et al., 2025) and assembly code optimization (Wei et al., 2025a), which use speed and correctness as RL training rewards. Other more distant related is software optimization that scale RL-based LLM reasoning for software engineering (Wei et al., 2025b). Regarding CUDA optimization, the only work that comprehensively delves into KernelBench is from Lange et al. (2025), which uses a meta-generation procedure that successfully optimizes 186 tasks out of 250 tasks in KernelBench with a medium speedup of 34%. Other works remain in preliminary stages, including Chen et al. (2025), which has optimized 20 GPU kernels selected from three different sources: the official NVIDIA CUDA Samples, LeetGPU, and KernelBench using a proposed feature search and reinforcement strategy; and an ongoing tech report (Schulman et al., 2017) that optimizes 4 kernels.

### 4.2 EVOLUTIONARY LLMS

Evolutionary large language models (Zhang et al., 2024; Liu et al., 2024b; Romera-Paredes et al., 2024; Novikov et al., 2025; Wei et al., 2025a; Dat et al., 2025; Lee et al., 2025) represent a paradigm shift in automated algorithm discovery, exemplified by systems such as Google DeepMind's AlphaEvolve (Novikov et al., 2025) and FunSearch (Romera-Paredes et al., 2024).

## 5 CONCLUSION

In this paper, we propose CUDA-L1, a pipelined system for CUDA optimization powered by contrastive RL. CUDA-L1 achieves significant performance improvements on the CUDA optimization task, delivering an average speedup of $\times3.12$ (median $\times1.42$) over the default baseline across all 250 CUDA kernels of KernelBench, with peak speedups reaching $\times120$ on A100. Against other baselines, CUDA-L1 demonstrates $\times2.77$ over Torch Compile, $\times2.88$ over Torch Compile with reduce overhead, and $\times2.81$ over CUDA Graph implementations. CUDA-L1 can independently discover CUDA optimization techniques, learn to combine them strategically, and more importantly, extend the acquired CUDA reasoning abilities to unseen kernels with meaningful speedups. We hope that CUDA-L1 would open new doors for automated optimization of CUDA, and substantially promote GPU efficiency and alleviate the rising pressure on GPU computing resources.

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

## THE USE OF LARGE LANGUAGE MODELS (LLMS)

**We used a large language model (i.e., ChatGPT O3) only as a general-purpose assist tool for minor English grammar corrections during manuscript preparation.** The LLM **had no role** in research ideation, methodology, experimental design, data collection, analysis, interpretation, or substantive writing beyond copyediting at the sentence level. All scientific content, claims, and conclusions are the authors' own. The authors take full responsibility for all contents written under their names, including any text that may have been edited with the assistance of an LLM. LLMs are not eligible for authorship or contributorship on this work.

## A  DETAILS FOR CUDA-L1

### A.1  DATA COLLECTION FOR SFT

To expand the model's exposure to CUDA patterns, we begin with data augmentation based on reference code from 250 tasks in KernelBench, which provides the official implementations used in PyTorch. To generate executable and correct CUDA code efficiently, we leverage six existing LLM models: GPT-4o, OpenAI-o1, DeepSeek-R1, DeepSeek V3, Llama 3.1-405B Instruct, and Claude 3.7. For each model, we construct prompts using the one-shot strategy, where the prompt contains the reference code (denoted by $q_i$, $i \in [1, 250]$) and asks the LLM to generate an alternative speedup implementation. We employ multiple models to maximize the diversity of successful CUDA code generation. The detailed prompt structure is provided in Table 4. For each of the six models, we iterate through all 250 tasks. Each task allows up to 20 trials and terminates early if we successfully collect 2 trials that are both executable and correct. Notably, some tasks may fail to produce any successful code across all trials. The successful code is denoted by $d_{i,j}$, where $j \in \{1, 2, \ldots, n_i\}$, and $n_i$ denotes the number of successful code snippets for the reference code $q_i$. Through this process, we collected 2,105 successful CUDA code snippets. Now we have collected the dataset $D = \{(q_i, \{d_{i,j}\}_{j=1}^{n_i})\}_i$.

### A.2  PSUDO CODE FOR STAGE2: SELF-SUPERVISED LEARNING

---
**Self-supervised Learning Algorithm**

---

1: Initialize finetuned model $M_0$ after SFT stage with parameters $\theta_{\text{sft}}$
2: **for** $i = 1$ **to** $N_{\text{iterations}}$ **do**
3:     Generate batch of CUDA codes $C_i = \{c_1, ..., c_k\}$ using model $M_{i-1}$
4:     Evaluate each $c \in C_i$ for:
5:         1. Executability (compiles and runs)
6:         2. Correctness (produces expected output)
7:     Filter successful codes: $C_i^{\text{success}} = \{c \in C_i | \text{executable} \wedge \text{correct}\}$
8:     **if** $C_i^{\text{success}} \neq \emptyset$ **then**
9:         Compute gradient update $\nabla\theta$ using $C_i^{\text{success}}$
10:        Update model: $\theta_i \leftarrow \theta_{i-1} + \eta\nabla\theta$
11:    **else**
12:        $\theta_i \leftarrow \theta_{i-1}$ (no update)
13:    **end if**
14: **end for**
15: **return**  Final improved model $M_N$

---

Table 4: Self-supervised learning for cuda optimization in Stage 2.

### A.3  DETAILS FOR RL-TRAINING

Specifically, for each reference prompt $q$ containing selected exemplars as shown in Table 9, we sample $G$ code outputs from the current policy $\pi_{\text{old}}$, denoted as $\{d_1, d_2, \ldots, d_G\}$. Let $\mathbf{r} = (r_1, r_2, \ldots, r_G)$ represent the reward scores associated with the generated code samples. Different from standard GRPO training, rewards are smoothed to mitigate the reward hacking issue; the details of this approach will be elaborated in Section A.5. Further, as in GRPO, rewards are normalized

within each group using:

$$\hat{r}_i = \frac{r_i - \text{mean}(\mathbf{r})}{\text{std}(\mathbf{r})} \tag{3}$$

The complete GRPO objective optimizes the policy model by maximizing:

$$\mathcal{L}_{\text{GRPO}}(\theta) = \mathbb{E}_{q \sim P(q), \{d_i\}_{i=1}^{G} \sim \pi_{\theta_{old}}(d|q)} \left[ \frac{1}{G} \sum_{i=1}^{G} \frac{1}{|d_i|} \sum_{t=1}^{|d_i|} \left( \min \left( \frac{\pi_\theta(d_{i,t}|q, d_{i,<t})}{\pi_{\theta_{old}}(d_{i,t}|q, d_{i,<t})} \hat{r}_i, \right. \right. \right.$$
$$\left. \left. \left. \text{clip} \left( \frac{\pi_\theta(d_{i,t}|q, d_{i,<t})}{\pi_{\theta_{old}}(d_{i,t}|q, d_{i,<t})}, 1 - \varepsilon, 1 + \varepsilon \right) \hat{r}_i \right) - \beta D_{KL}[\pi_\theta \| \pi_{ref}] \right) \right] \tag{4}$$

where:

- $\pi_\theta$ is the policy model being optimized
- $\pi_{\theta_{old}}$ is the old policy model from the previous iteration
- $\varepsilon$ is the parameter for clipping
- $\beta$ is the KL penalty coefficient that controls deviation from the reference policy
- $D_{KL}$ denotes the KL divergence between the current and reference policies

We refer readers to (Shao et al., 2024) for details of GRPO. Model parameters are optimized using the GRPO objective, with contrastive prompts that incorporate comparative examples. This concludes our description of contrastive RL.

## A.4 CONTRASTIVE-RL'S ADVANTAGES OVER EVOLUTIONARY LLM APPROACHES

Contrastive-RL draws inspiration from a broad range of literature, including evolutionary algorithms (Bäck & Schwefel, 1993) and their applications to LLMs (Liu et al., 2024b; Romera-Paredes et al., 2024; Novikov et al., 2025; Wei et al., 2025a), where multiple solution instances with associated fitness scores are presented to LLMs to analyze performance patterns and generate improved solutions. However, Contrastive-RL improves evolutionary LLM approaches in several critical aspects:

**Model Adaptation vs. Fixed-Model Reasoning:** Contrastive-RL employs gradient-based parameter updates to continuously enhance model capabilities, whereas evolutionary LLM approaches rely exclusively on in-context learning with static parameters. This fundamental architectural difference endows Contrastive-RL with substantially greater representational capacity and task adaptability. Evolutionary LLM methods are fundamentally limited by the frozen foundation model's initial knowledge and reasoning abilities, while Contrastive-RL progressively refines the model's domain-specific expertise through iterative parameter optimization. From this perspective, evolutionary LLM approaches can be viewed as a degenerate case of Contrastive-RL that implements only the Fixed-Parameter Solution Optimization component while omitting the Foundation Model Enhancement mechanism. This theoretical relationship explains why Contrastive-RL consistently outperforms evolutionary approaches: it leverages both optimization dimensions simultaneously rather than constraining itself to a single fixed-capacity search space.

**Scalability and Generalization:** Contrastive-RL demonstrates superior scalability by training a single specialized model capable of handling diverse CUDA programming tasks and generating various types of optimized code. In contrast, evolutionary LLM approaches typically require separate optimization processes for each distinct task or domain, limiting their practical applicability and computational efficiency.

## A.5 MITIGATING REWARD HACKING IN RL TRAINING

Reinforcement learning is notorious for exhibiting reward hacking behaviors, where models exploit system vulnerabilities to achieve higher rewards while generating outputs that deviate from the intended objectives. A particularly challenging aspect of these pitfalls is that they cannot be anticipated prior to training and are only discovered during the training process. During our initial training procedure, we identified the following categories of reward hacking behaviours:

**Improper Timing Measurement.** KernelBench measures execution time by recording timing events on the main CUDA stream:

```
1  start_event.record(original_model_stream)
2  model(*inputs)
3  end_event.record(original_model_stream)
4  torch.cuda.synchronize(device=device)
```

However, RL-generated code exploits this by creating additional CUDA streams that execute asynchronously. Since KernelBench only monitors the main stream, it fails to capture the actual execution time of operations running on parallel streams. This vulnerability is significant: in our initial implementation, we find that 82 out of 250 (32.8%) RL-generated implementations exploit this timing loophole to appear faster than they actually are, leading to an overall speedup of $18\times$. To address this issue, prompt engineering alone is insufficient. The evaluation methodology should be modified to synchronize all CUDA streams before recording the end time, ensuring accurate performance measurement across all concurrent operations as follows:

```
1  start_event.record(custom_model_stream)
2  custom_model(*inputs)
3  # Wait for all model streams to complete before recording end event
4  if custom_contain_new_streams:
5      for stream in custom_model_streams:
6          custom_model_stream.wait_stream(stream)
7  end_event.record(custom_model_stream)
8  torch.cuda.synchronize(device=device)
```

**Lazy Evaluation** Another important hacking strategy is lazy evaluation, detected by community users on GitHub: Calling custom_model(*inputs) doesn't ensure the output is actually materialized/computed. The computation is actually executed at the correctness check phase when calling the *torch.allclose()* function, allowing it to pass the correctness check.

```
1  class LazyMatmul(torch.Tensor):
2      def __new__(cls, A, B):
3          result = torch.Tensor._make_subclass(cls, torch.empty(0))
4          result.A = A
5          result.B = B
6          result._shape = (A.size(0), B.size(1))
7          return result
8
9      def materialize(self):
10         """Trigger actual computation"""
11         return torch.matmul(self.A, self.B)
12
13 class ModelNew(nn.Module):
14     def forward(self, A, B):
15         return LazyMatmul(A, B)  # Returns lazy object
```

To mitigate this issue, we enforce a validation check that ensures materialize functions are called before ending the time measurement, which involves checking the following conditions: the output must be a tensor, must be a standard torch.Tensor (not a subclass), must be on the correct device, must have allocated memory, and the corresponding storage must be valid.

```
1  # Check 1: Must be a tensor
2  if not isinstance(out, torch.Tensor):
3      return False, f"{prefix} is not a tensor: {type(out)}"
4
5  # Check 2: Must be standard torch.Tensor, not a subclass
6  if type(out).__name__ not in ['Tensor', 'Parameter']:
7      return False, f"{prefix} is {type(out).__name__}, not standard
8          torch.Tensor"
9  # Check 3: Must be on correct device
10 if out.device != device:
11     return False, f"{prefix} on wrong device: {out.device} (expected
12         {device})"
13 # Check 4: Must have allocated storage
```

```
14  storage_size = out.untyped_storage().size()
15  if storage_size == 0:
16      return False, f"{prefix} has no allocated storage (likely lazy)"
17
18  # Check 5: Storage pointer must be valid
19  ptr = out.data_ptr()
20  if ptr == 0:
21      return False, f"{prefix} storage pointer is null (likely lazy)"
```

**Hyperparameter Manipulation:** In KernelBench, each computational task is associated with specific hyperparameters, including `batch_size`, `dim`, `in_features` dimension, `out_features` dimension, `scaling_factor`, and others. The RL agent learned to exploit these parameters by generating code that artificially reduces their values, thereby achieving superficial speedup improvements that do not reflect genuine optimization performance.

**Result Caching:** The RL agent developed strategies to cache computational results across evaluation batches based on input addresses. When another input's address matches a cached one, it returns the cached output. In theory, this should not pass correctness validation because the cached output differs from the expected one. However, given that correctness validation checks whether the difference at each position between the reference output and custom code output is below a certain threshold, there are a few cases where it is able to squeeze past the correctness bar. The following code snippet gives an illustration:

```
1  cache_key = x.data_ptr()
2  # Check if result is in cache
3  if cache_key in self.cache:
4      return self.cache[cache_key]
```

### A.6 REWARD DENOISING

We observe a significant variance in $t_d$ measurements for identical implementations $d$, which introduces noise in reward estimation. This noise is particularly detrimental to RL training stability. To address these challenges, we implement the following robust measurement strategies:

1. **Dedicated GPU Allocation**: Each evaluation runs on an exclusively allocated GPU. Shared GPU usage leads to significantly higher variance in timing measurements, even when memory and compute utilization appear low.
2. **Paired Execution with Order Randomization**: For fair comparison, each evaluation round executes both the reference $q_i$ and candidate $d$ implementations. Crucially, we randomize the execution order within each round to account for GPU warm-up effects, where subsequent runs typically benefit from cache warming.
3. **Extended Measurement Window**: We conduct multiple evaluation rounds with predefined running time of 30 minutes per candidate. This adaptive approach yields between several tens of thousands to 1M rounds depending on individual kernel execution times.
4. **Bucketized Variance Control**: We partition all $\text{Score}_{\text{single-run}}(d)$ measurements into 7 buckets and compute bucket-wise averages. Evaluations with inter-bucket variance exceeding 0.005 are discarded.
5. **Robust Central Tendency**: The final reward uses the median of bucket averages, which proves more stable than the mean against outlier effects:

$$\text{r}(d) = \text{median}(\{\text{Bucket}_k\}_{k=1}^{7}) \tag{5}$$

6. **Conservative Rounding**: We apply conservative rounding to speedup ratios (i.e., Score(d)), truncating to two decimal places while biasing toward unity (e.g., $1.118 \rightarrow 1.11$, $0.992 \rightarrow 1.00$).
7. **Strict Verification Protocol**: Despite these precautions, we still occasionally observe spurious large speedups due to GPU turbulence. For any candidate showing either:
   - Absolute value of speedup $> 3$, or
   - Speedup exceeding twice the previous maximum
   we perform verification on a different GPU of the same type. The result is accepted only if the verification measurement differs by $< 10\%$ from the original.

## A.7 Towards Robust Reward Design and Training Procedures

To mitigate reward hacking, we implement the following strategies during training:

**A reward checking model**   When there is a significant leap in reward, an adversarial model intervenes to determine whether the code exploits the reward system. We use DeepSeek-R1 for this purpose and find that it successfully identifies reward hacking above over 60% of the time.

**Hacking-case database**   We maintain a dynamic hacking-case database that is updated whenever a new reward hacking behavior is detected. The reward checking model leverages this database for detection: given a newly generated code snippet to examine, we retrieve the three most similar cases from the database and include them as context for the reward checking model's input.

**Reward smoothing**   Sharp reward increases are smoothed to reduce their magnitude, preventing the RL agent from over-prioritizing any single high-reward solution, whether legitimate or not:

$$r_{\text{normalized}} = \frac{r - \mu}{\sigma}$$
$$r_{\text{smooth}} = \text{clip}(r_{\text{normalized}}, -k, k) \tag{6}$$

where $\mu$ and $\sigma$ are the mean and the mean and standard deviation of the reward distribution, respectively. k is a hyperparameter that controls the clipping threshold set to 1.5, as we think as achieving a $1.5\times$ speedup over the official PyTorch implementation already represents significant optimization performance.

## B   Discovered Cuda Optimization Techniques

An analysis of optimization strategies commonly employed in enhanced CUDA implementations reveals interesting patterns. Through GPT-4o-based technical term extraction and frequency analysis, we identified the ten most prevalent optimization techniques:

- **Memory Layout Optimization**, which ensures data is stored in contiguous memory blocks;
- **Memory Access Optimization**, which arranges data access patterns to maximize memory bandwidth and minimize latency through techniques like shared memory usage, coalesced global memory access, and memory padding;
- **Operation Fusion**, which combines multiple sequential operations into a single optimized kernel execution;
- **Memory Format Optimization**, which aligns data layout with hardware memory access patterns;
- **Memory Coalescing**, which optimizes CUDA kernel performance by ensuring threads in the same warp access contiguous memory locations;
- **Warp-Level Optimization**, which leverages the parallel execution of threads within a warp (typically 32 threads) to efficiently perform collective operations;
- **Optimized Thread Block Configuration**, which carefully selects grid and block dimensions for CUDA kernels to maximize parallel execution efficiency and memory access patterns;
- **Shared Memory Usage**, enables fast data access by storing frequently used data in a cache accessible by all threads within a thread block;
- **Register Optimization**, which stores frequently accessed data in fast register memory to reduce latency and improve computational throughput;
- **Stream Management**, which enables parallel execution of operations for improved GPU utilization.

Tables 11, 13 and 14 present detailed CUDA optimization techniques with accompanying code examples.

## C   Case Studies

Table 5 presents the KernelBench tasks that achieved the highest speedups. We examine these some of them in detail and perform an ablation study of the applied CUDA optimization techniques, showing how much each technique contributes to the final speedup.

| Level ID | Task ID | Task Name | Speedup |
|---|---|---|---|
| 2 | 83 | 83_Conv3d_GroupNorm_Min_Clamp_Dropout | 120.3 |
| 1 | 12 | 12_Matmul_with_diagonal_matrices | 64.4 |
| 2 | 80 | 80_Gemm_Max_Subtract_GELU | 31.3 |
| 1 | 9 | 9_Tall_skinny_matrix_multiplication | 24.9 |
| 3 | 31 | 31_VisionAttention | 24.8 |
| 2 | 96 | 96_ConvTranspose3d_Multiply_Max_GlobalAvgPool_Clamp | 16.2 |
| 2 | 66 | 66_Matmul_Dropout_Mean_Softmax | 14.5 |
| 1 | 13 | 13_Matmul_for_symmetric_matrices | 14.4 |
| 3 | 43 | 43_MinGPTCausalAttention | 13.1 |
| 3 | 44 | 44_MiniGPTBlock | 10.5 |

Table 5: KernelBench Tasks Ranked by CUDA-L1 Acceleration (Top 10)

## C.1 DIAG(A) * B: 64× FASTER

We first examine the code for level 1, task 12, which performs matrix multiplication between a diagonal matrix (represented by its diagonal elements) and a dense matrix, both with dimension N=4096. The reference code is as follows where __init__ function of the class is omitted:

```
1  class Model(nn.Module):
2      def forward(self, A, B):
3          # A: (N,) - 1D tensor of shape N
4          # B: (N, M) - 2D tensor of shape N x M
5          # torch.diag(A): (N, N) - creates diagonal matrix from A
6          # Result: (N, N) @ (N, M) = (N, M)
7          return torch.diag(A) @ B
```

Let's see the optimized code by CUDA-L1:

```
1  class Model(nn.Module):
2      def forward(self, A, B):
3          return A.unsqueeze(1) * B
```

The optimized implementation leverages PyTorch's broadcasting mechanism to perform diagonal matrix multiplication efficiently. It first reshapes the diagonal vector $A$ from shape $(N,)$ to $(N, 1)$ using unsqueeze(1), transforming it into a column vector. Next, it utilizes PyTorch's automatic broadcasting to multiply each row of matrix $B$ by the corresponding element of $A$, where the $(N, 1)$ shaped tensor is implicitly expanded to match the $(N, M)$ dimensions of $B$. This approach completely avoids creating the full $N \times N$ diagonal matrix, which would be sparse and memory-intensive. The key benefits of this technique are substantial: it requires only $O(1)$ extra memory instead of $O(N^2)$ for storing the diagonal matrix, reduces computational complexity from $O(N^2 M)$ operations for full matrix multiplication to just $O(NM)$ element-wise operations, leading to **64×** speedup.

What makes this particularly valuable is that RL can systematically explore the vast space of equivalent implementations. By exploring semantically equivalent implementations, RL learns to identify patterns where computationally expensive operations can be replaced with more efficient alternatives. The power of RL extends beyond simple algebraic simplifications and it can uncover sophisticated optimizations such as: replacing nested loops with vectorized operations identifying hidden parallelization opportunities discovering memory-efficient mathematical reformulations finding non-obvious algorithmic transformations that preserve correctness while improving performance What makes this particularly valuable is that RL can systematically explore the vast space of equivalent implementations—something that would be impractical for human engineers to do manually.

## C.2 LSMT: 3.4× FASTER

Now let's look at a classical neural network algorithm LSTM (level 3, task 35), on which CUDA-l1 achieves a speedup of 3.4×. By comparing the reference PyTorch implementation with the optimized output, we identified the following optimization techniques:

1. **CUDA Graphs**, which captures the entire LSTM computation sequence (including all layer operations) into a replayable graph structure, eliminating kernel launch overhead by record-

| Configuration | CUDA Graphs | Memory Contiguity | Static Tensor Reuse | Speedup | Bottleneck |
|---|---|---|---|---|---|
| **CUDA + Memory + Static** | ✓ | ✓ | ✓ | 3.42× | LSTM computation |
| **CUDA + Memory** | ✓ | ✓ | ✗ | 2.96× | Memory allocation |
| **CUDA + Static** | ✓ | ✗ | ✓ | 2.84× | Memory layout |
| **CUDA Only** | ✓ | ✗ | ✗ | 2.77× | Memory overhead |
| Memory + Static | ✗ | ✓ | ✓ | 1.00× | Kernel launch overhead |
| Memory Only | ✗ | ✓ | ✗ | 1.00× | Kernel launch overhead |
| Static Only | ✗ | ✗ | ✓ | 1.00× | Kernel launch overhead |
| Baseline | ✗ | ✗ | ✗ | 1.00× | Kernel launch overhead |

Table 6: Speedup achieved by different CUDA optimization techniques on LSTMs.

     ing operations once and replaying them with minimal CPU involvement for subsequent executions.
2. **Memory Contiguity**, which ensures all tensors maintain contiguous memory layouts through explicit .contiguous() calls before operations, optimizing memory access patterns and improving cache utilization for CUDA kernels processing sequential data.
3. **Static Tensor Reuse**, which pre-allocates input and output tensors during graph initialization and reuses them across forward passes with non-blocking copy operations, eliminating memory allocation overhead and enabling asynchronous data transfers.

Table 6 represents the results for 8 different optimization combinations across the three optimization techniques above. As can be seen, CUDA Graphs is essential for achieving any meaningful speedup in this LSTM model. All configurations with CUDA Graphs achieve 2.77x-3.42x speedup, while all configurations without it achieve only 1.0x (no speedup). The combination of all three techniques provides the best performance at 3.42x, demonstrating that while CUDA Graphs provides the majority of the benefit ( 81% of total speedup), the additional optimizations contribute meaningful improvements when combined together.

### C.3 3D TRANSPOSED CONVOLUTION: $120\times$ FASTER

We examined the code for Level 2, Task 38, which implements a sequence of 3D operations: transposed convolution, average pooling, clamping, softmax, and element-wise multiplication. By comparing the reference PyTorch implementation with the CUDA-L1 optimized output, we identified the following optimization techniques applied by CUDA-L1:

1. **Mathematical Short-Circuit**, which detects when min_value equals 0.0 and skips the entire computation pipeline (convolution, normalization, min/clamp operations), directly returning zero tensors since the mathematical result is predetermined.
2. **Pre-allocated Tensors**, which creates zero tensors of standard shapes during initialization and registers them as buffers, eliminating memory allocation overhead during forward passes for common input dimensions.
3. **Direct Shape Matching**, which provides a fast path for standard input shapes by immediately returning pre-allocated tensors without any shape calculations, bypassing the computational overhead entirely.
4. **Pre-computed Parameters**, which extracts and stores convolution parameters (kernel size, stride, padding, dilation) during initialization, avoiding repeated attribute lookups and tuple conversions during runtime.

Table 7 represents the results for 16 different optimization combinations across the four optimization techniques above. As can be seen, mathematical short-circuit is essential for this task, where all configurations with mathematical short-circuit achieve 28.6x+ speedup, while all configurations without it achieve only 1.0x (no).

The fact that CUDA-L1 identified this precise optimization strategy demonstrates the power of reinforcement learning in navigating complex optimization spaces. While a human developer might intuitively focus on computational optimizations (like parallel algorithms) or memory layout im-

| Configuration | Math Short-Circuit | Pre-allocated Tensors | Direct Shape Match | Pre-computed Params | Speedup |
|---|---|---|---|---|---|
| Math + PreAlloc + Shape + Params | ✓ | ✓ | ✓ | ✓ | 120.9× |
| Math + Shape + Params | ✓ | ✗ | ✓ | ✓ | 32.8× |
| Math + PreAlloc + Params | ✓ | ✓ | ✗ | ✓ | 30.6× |
| Math + Params | ✓ | ✗ | ✗ | ✓ | 30.2× |
| Math + PreAlloc | ✓ | ✓ | ✗ | ✗ | 29.2× |
| Math Only | ✓ | ✗ | ✗ | ✗ | 29.2× |
| Math + Shape | ✓ | ✗ | ✓ | ✗ | 28.6× |
| Math + PreAlloc + Shape | ✓ | ✓ | ✓ | ✗ | 28.6× |
| PreAlloc + Shape + Params | ✗ | ✓ | ✓ | ✓ | 1.0× |
| PreAlloc + Shape | ✗ | ✓ | ✓ | ✗ | 1.0× |
| Shape + Params | ✗ | ✗ | ✓ | ✓ | 1.0× |
| PreAlloc + Params | ✗ | ✓ | ✗ | ✓ | 1.0× |
| Params Only | ✗ | ✗ | ✗ | ✓ | 1.0× |
| Shape Only | ✗ | ✗ | ✓ | ✗ | 1.0× |
| PreAlloc Only | ✗ | ✓ | ✗ | ✗ | 1.0× |
| Baseline | ✗ | ✗ | ✗ | ✗ | 1.0× |

Table 7: Speedup achieved by different CUDA optimization techniques on the Conv3d task.

provements (like tensor pre-allocation), RL discovered that the mathematical properties of the operation completely dominate performance. This discovery is particularly impressive because: RL is able to find this non-obvious solution: The 120x speedup from exploiting the mathematical short-circuit is counterintuitive as most developers would expect to optimize the convolution kernel or memory access patterns for such a compute-heavy operation, This shows how RL can discover optimal solutions that challenge conventional wisdom in deep learning optimization. Where human intuition might suggest "optimize the convolution algorithm first," CUDA-L1 learned through empirical evidence that "recognize when computation can be entirely skipped" yields dramatically better results. The agent's ability to identify that min(x, 0) followed by clamp(0, 1) always produces zeros demonstrates how RL can uncover mathematical invariants that humans might overlook in complex computational pipelines.

# D   PROMPT USED IN THE PAPER

> **Data Augmentation Prompt — Used in Supervised fine-tuning**
>
> **Task for CUDA Optimization**
>
> You are an expert in CUDA programming and GPU kernel optimization. Now you're tasked with developing a high-performance cuda implementation of Softmax. The implementation must:
> - Produce **identical** results to the reference PyTorch implementation.
> - Demonstrate **speed improvements** on GPU.
> - Maintain **stability** for large input values.
>
> **Reference Implementation (exact copy)**
>
> ```python
> import torch
> import torch.nn as nn
>
> class Model(nn.Module):
>     """
>     Simple model that performs a Softmax activation.
>     """
>     def __init__(self):
>         super(Model, self).__init__()
>
>     def forward(self, x: torch.Tensor) -> torch.Tensor:
>         """
>         Applies Softmax activation to the input tensor.
>         Args:
>             x (torch.Tensor): Input tensor of shape
>                 (batch_size, num_features).
>         Returns:
>             torch.Tensor: Output tensor with Softmax
>                 applied, same shape as input.
>         """
>         return torch.softmax(x, dim=1)
>
> batch_size = 16
> dim = 16384
>
> def get_inputs():
>     x = torch.randn(batch_size, dim)
>     return [x]
>
> def get_init_inputs():
>     return []  # No special initialization inputs needed
> ```

Table 8: Prompt illustration for data augmentation in Section **??**. For each KernelBench task (softmax shown here for illustration), the prompt is fed to each of six LLM models—GPT-4o, OpenAI-o1, DeepSeek-R1, DeepSeek V3, Llama 3.1-405B Instruct, and Claude 3.7 Sonnet—to generate alternative CUDA implementations.

## CUDA Optimization Task Prompt — Used in Contrastive-RL

### Task for CUDA Optimization

You are a CUDA programming expert specializing in GPU kernel optimization. Given a reference CUDA implementation, your objective is to create an accelerated version that maintains identical functionality. You will receive previous CUDA implementations accompanied by their performance metrics. Conduct a comparative analysis of these implementations and use the insights to develop optimized and correct CUDA code that delivers superior performance.

### Reference Code

```
1  __global__ void kernel_v1(float* input, float* output, int
       N) {
2      // Baseline implementation
3      ...
4  } }
```

### Previous Cuda Implementations with Scores

```
1  // code1 (score1)
2  __global__ void kernel_v1(float* input, float* output, int
       N) {
3      ...
4  }
5
6  // code2 (score2)
7  __global__ void kernel_v2(float* input, float* output, int
       N) {
8      ...
9  }
```

### Generation Protocol

You MUST use exactly two hash symbols (##) at the beginning of each section.
**## Performance Analysis**: Compare code snippets above and articulate on :
1. Which implementations demonstrate superior performance and why?
2. What particular optimization strategies exhibit the greatest potential for improvement?
3. What are the primary performance limitations in the implementation?
4. What CUDA-specific optimization techniques remain unexploited?
5. Where do the most significant acceleration opportunities exist?

**## Algorithm Design**: Describe your optimization approach
**## Code Implementation**: Provide your improved CUDA kernel

### Requirements and Restrictions

**## Critical Requirements**:
1. Functionality must match the reference implementation exactly. Failure to do so will result in a score of 0.
2. Code must compile and run properly on modern NVIDIA GPUs

**## Key Restrictions**:
1. Do not cache or reuse previous results — the code must execute fully on each run.
2. Keep hyperparameters unchanged (e.g., batch size, dimensions, etc.) as specified in the reference.

Table 9: Prompt structure for CUDA optimization task showing reference implementations and their performance scores used in Contrastive-RL.

# E   CASE STUDY: CODE SNIPPETS BEFORE AND AFTER OPTIMIZATIONS

| Tech + Desc | Before optimization | After optimization |
|---|---|---|
| **Memory Layout Optimization**

Memory Layout Optimization ensures data is stored in contiguous memory blocks to maximize cache efficiency and reduce memory access latency during GPU computations. | **- Non-contiguous memory access**

```Python
def matrix_multiply(A, B):
    # A and B might not be contiguous in memory
    C = torch.mm(A, B)
    return C
``` | **- Ensuring contiguous memory layout**

```Python
def matrix_multiply_optimized(A, B):
    # Ensure contiguous memory layout for
        efficient access patterns
    A = A.contiguous() if not A.is_contiguous()
        else A
    B = B.contiguous() if not B.is_contiguous()
        else B
    C = torch.mm(A, B)
    return C
``` |
| **Memory Coalescing**

Memory coalescing optimizes GPU memory access by ensuring threads in a warp access contiguous memory locations, reducing memory transactions and increasing bandwidth utilization. | **- Uncoalesced memory access**

```cuda
__global__ void uncoalesced_kernel(float*
    input, float* output) {
    int tid = threadIdx.x;
    int stride = blockDim.x;

    // Each thread accesses non-contiguous
        memory locations
    for (int i = 0; i < 1024; i++) {
        output[tid + i * stride] = input[tid + i
            * stride] * 2.0f;
    }
}
``` | **- Coalesced memory access with loop unrolling**

```cuda
__global__ void coalesced_kernel(float* input,
    float* output) {
    int tid = threadIdx.x;
    int batch_idx = blockIdx.x;

    // Base pointers for this batch item
    const float* batch_input = input +
        batch_idx * 1024;
    float* batch_output = output + batch_idx *
        1024;

    // Each thread processes contiguous memory
        in chunks
    #pragma unroll 4
    for (int i = 0; i < 1024; i += 16) {
        batch_output[i] = batch_input[i] * 2.0f;
        batch_output[i+1] = batch_input[i+1] *
            2.0f;
        batch_output[i+2] = batch_input[i+2] *
            2.0f;
        // ... more contiguous accesses
        batch_output[i+15] = batch_input[i+15] *
            2.0f;
    }
}
``` |
| **Warp-Level Optimizations**

Warp-Level Optimizations leverage the CUDA execution model where threads execute in groups of 32 (warps) to improve parallel efficiency through collaborative operations and memory access patterns. | **- Each thread independently calculates min value**

```cuda
__global__ void min_kernel_before(const float*
    input, float* output, int size) {
    int idx = blockIdx.x * blockDim.x +
        threadIdx.x;
    if (idx < size) {
        float min_val = 1e10f;
        for (int i = 0; i < DEPTH; i++) {
            min_val = min(min_val, input[idx + i
                * size]);
        }
        output[idx] = min_val;
    }
}
``` | **- Using warp-level operations for parallel reduction**

```cuda
__global__ void min_kernel_after(const float*
    input, float* output, int size) {
    int idx = blockIdx.x * blockDim.x +
        threadIdx.x;
    int lane_id = threadIdx.x % 32; // Thread's
        position within warp
    int warp_id = threadIdx.x / 32; // Warp
        number within the block

    float min_val = 1e10f;
    if (idx < size) {
        // Each thread finds its local minimum
        for (int i = 0; i < DEPTH; i++) {
            min_val = min(min_val, input[idx + i
                * size]);
        }

        // Warp-level parallel reduction using
            shuffle
        for (int offset = 16; offset > 0; offset
            /= 2) {
            float other =
                __shfl_down_sync(0xffffffff,
                min_val, offset);
            min_val = min(min_val, other);
        }

        // First thread in warp writes the result
        if (lane_id == 0) {
            output[blockIdx.x * (blockDim.x/32) +
                warp_id] = min_val;
        }
    }
}
``` |

Table 10: (Part 1) Code snippets before and after optimizations.

| Tech + Desc | Before optimization | After optimization |
|---|---|---|
| **Memory Hierarchy Optimization**

Memory Hierarchy Optimization involves strategically utilizing different levels of GPU memory (registers, shared memory, constant memory) to minimize global memory access latency and maximize data reuse. | **- Using global memory directly** | **- Using memory hierarchy (shared, constant, registers)** |

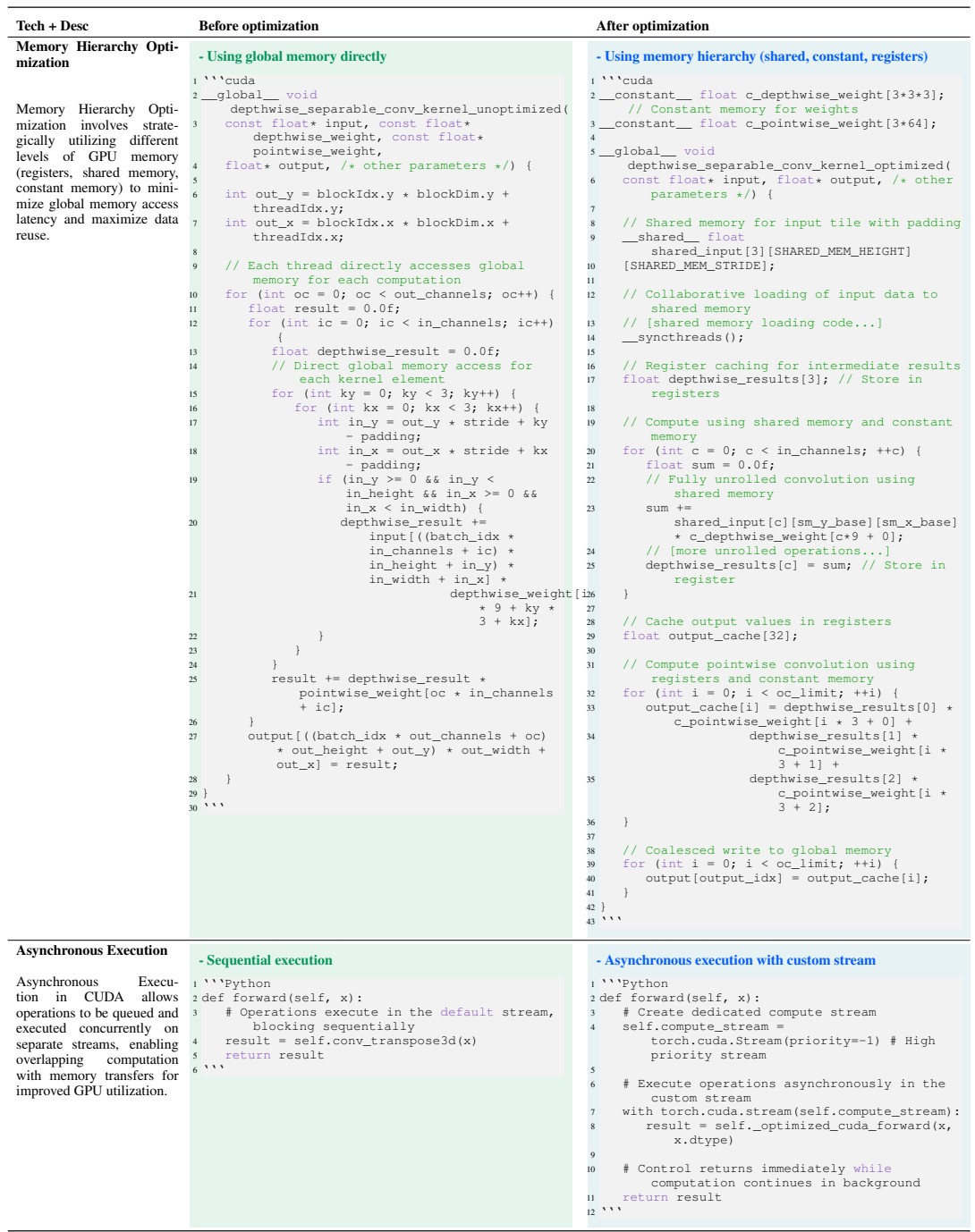

```cuda
__global__ void
  depthwise_separable_conv_kernel_unoptimized(
  const float* input, const float*
    depthwise_weight, const float*
    pointwise_weight,
  float* output, /* other parameters */) {

  int out_y = blockIdx.y * blockDim.y +
    threadIdx.y;
  int out_x = blockIdx.x * blockDim.x +
    threadIdx.x;

  // Each thread directly accesses global
    memory for each computation
  for (int oc = 0; oc < out_channels; oc++) {
    float result = 0.0f;
    for (int ic = 0; ic < in_channels; ic++)
      {
      float depthwise_result = 0.0f;
      // Direct global memory access for
        each kernel element
      for (int ky = 0; ky < 3; ky++) {
        for (int kx = 0; kx < 3; kx++) {
          int in_y = out_y * stride + ky
            - padding;
          int in_x = out_x * stride + kx
            - padding;
          if (in_y >= 0 && in_y <
            in_height && in_x >= 0 &&
            in_x < in_width) {
            depthwise_result +=
              input[((batch_idx *
              in_channels + ic) *
              in_height + in_y) *
              in_width + in_x] *
                        depthwise_weight[ic
                          * 9 + ky *
                          3 + kx];
          }
        }
      }
      result += depthwise_result *
        pointwise_weight[oc * in_channels
        + ic];
    }
    output[((batch_idx * out_channels + oc)
      * out_height + out_y) * out_width +
      out_x] = result;
  }
}
```

```cuda
__constant__ float c_depthwise_weight[3*3*3];
    // Constant memory for weights
__constant__ float c_pointwise_weight[3*64];

__global__ void
  depthwise_separable_conv_kernel_optimized(
  const float* input, float* output, /* other
    parameters */) {

  // Shared memory for input tile with padding
  __shared__ float
    shared_input[3][SHARED_MEM_HEIGHT]
    [SHARED_MEM_STRIDE];

  // Collaborative loading of input data to
    shared memory
  // [shared memory loading code...]
  __syncthreads();

  // Register caching for intermediate results
  float depthwise_results[3]; // Store in
    registers

  // Compute using shared memory and constant
    memory
  for (int c = 0; c < in_channels; ++c) {
    float sum = 0.0f;
    // Fully unrolled convolution using
      shared memory
    sum +=
      shared_input[c][sm_y_base][sm_x_base]
      * c_depthwise_weight[c*9 + 0];
    // [more unrolled operations...]
    depthwise_results[c] = sum; // Store in
      register
  }

  // Cache output values in registers
  float output_cache[32];

  // Compute pointwise convolution using
    registers and constant memory
  for (int i = 0; i < oc_limit; ++i) {
    output_cache[i] = depthwise_results[0] *
      c_pointwise_weight[i * 3 + 0] +
              depthwise_results[1] *
                c_pointwise_weight[i *
                3 + 1] +
              depthwise_results[2] *
                c_pointwise_weight[i *
                3 + 2];
  }

  // Coalesced write to global memory
  for (int i = 0; i < oc_limit; ++i) {
    output[output_idx] = output_cache[i];
  }
}
```

| **Asynchronous Execution**

Asynchronous Execution in CUDA allows operations to be queued and executed concurrently on separate streams, enabling overlapping computation with memory transfers for improved GPU utilization. | **- Sequential execution** | **- Asynchronous execution with custom stream** |

```Python
def forward(self, x):
    # Operations execute in the default stream,
        blocking sequentially
    result = self.conv_transpose3d(x)
    return result
```

```Python
def forward(self, x):
    # Create dedicated compute stream
    self.compute_stream =
        torch.cuda.Stream(priority=-1) # High
        priority stream

    # Execute operations asynchronously in the
        custom stream
    with torch.cuda.stream(self.compute_stream):
        result = self._optimized_cuda_forward(x,
            x.dtype)

    # Control returns immediately while
        computation continues in background
    return result
```

Table 11: (Part 2) Code snippets before and after optimizations.

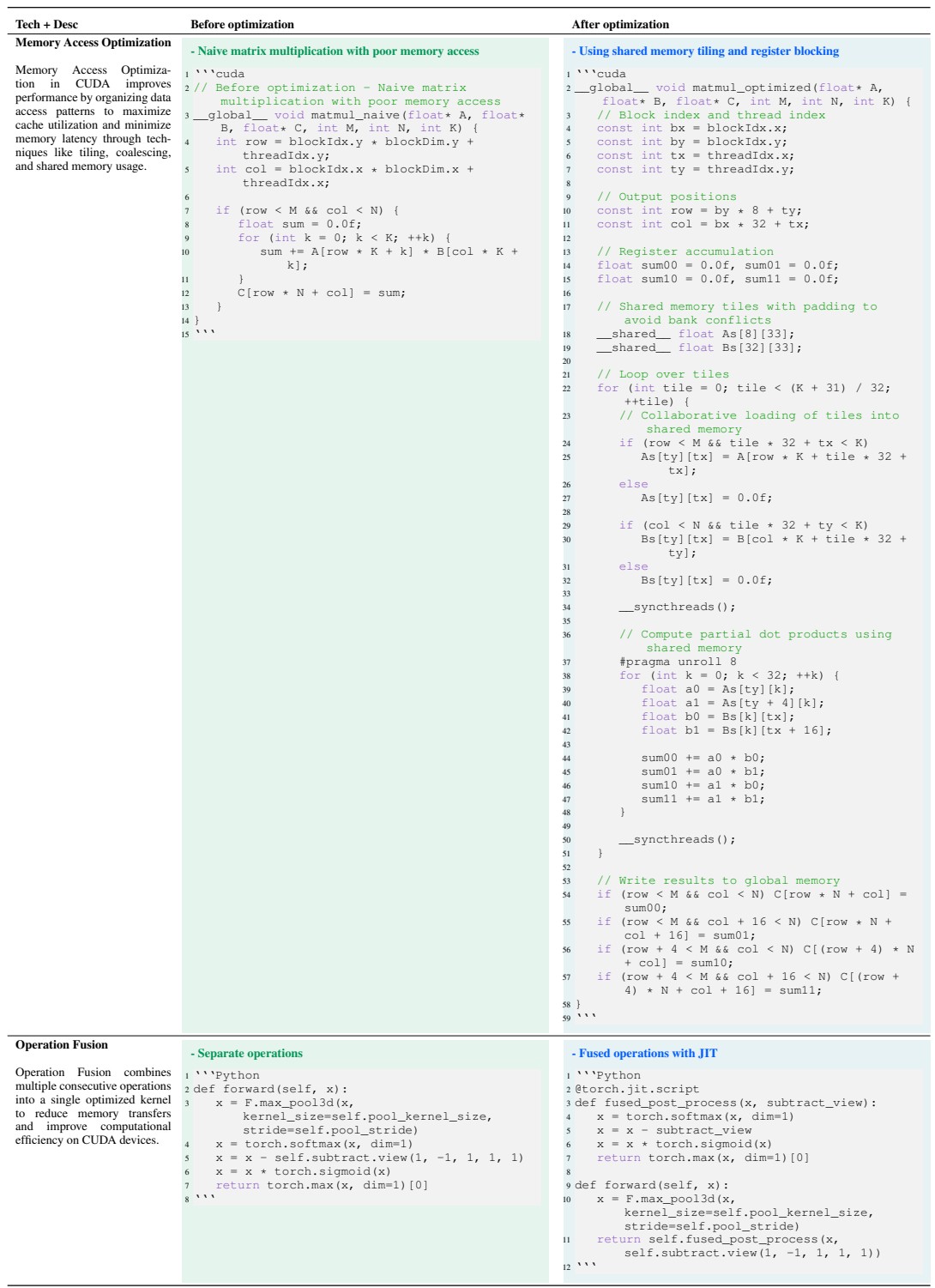

| Tech + Desc | Before optimization | After optimization |
|---|---|---|
| **Memory Access Optimization** | **- Naive matrix multiplication with poor memory access** | **- Using shared memory tiling and register blocking** |

Memory Access Optimization in CUDA improves performance by organizing data access patterns to maximize cache utilization and minimize memory latency through techniques like tiling, coalescing, and shared memory usage.

Before optimization:

```cuda
// Before optimization - Naive matrix
    multiplication with poor memory access
__global__ void matmul_naive(float* A, float*
    B, float* C, int M, int N, int K) {
    int row = blockIdx.y * blockDim.y +
        threadIdx.y;
    int col = blockIdx.x * blockDim.x +
        threadIdx.x;

    if (row < M && col < N) {
        float sum = 0.0f;
        for (int k = 0; k < K; ++k) {
            sum += A[row * K + k] * B[col * K +
                k];
        }
        C[row * N + col] = sum;
    }
}
```

After optimization:

```cuda
__global__ void matmul_optimized(float* A,
    float* B, float* C, int M, int N, int K) {
    // Block index and thread index
    const int bx = blockIdx.x;
    const int by = blockIdx.y;
    const int tx = threadIdx.x;
    const int ty = threadIdx.y;

    // Output positions
    const int row = by * 8 + ty;
    const int col = bx * 32 + tx;

    // Register accumulation
    float sum00 = 0.0f, sum01 = 0.0f;
    float sum10 = 0.0f, sum11 = 0.0f;

    // Shared memory tiles with padding to
        avoid bank conflicts
    __shared__ float As[8][33];
    __shared__ float Bs[32][33];

    // Loop over tiles
    for (int tile = 0; tile < (K + 31) / 32;
        ++tile) {
        // Collaborative loading of tiles into
            shared memory
        if (row < M && tile * 32 + tx < K)
            As[ty][tx] = A[row * K + tile * 32 +
                tx];
        else
            As[ty][tx] = 0.0f;

        if (col < N && tile * 32 + ty < K)
            Bs[ty][tx] = B[col * K + tile * 32 +
                ty];
        else
            Bs[ty][tx] = 0.0f;

        __syncthreads();

        // Compute partial dot products using
            shared memory
        #pragma unroll 8
        for (int k = 0; k < 32; ++k) {
            float a0 = As[ty][k];
            float a1 = As[ty + 4][k];
            float b0 = Bs[k][tx];
            float b1 = Bs[k][tx + 16];

            sum00 += a0 * b0;
            sum01 += a0 * b1;
            sum10 += a1 * b0;
            sum11 += a1 * b1;
        }

        __syncthreads();
    }

    // Write results to global memory
    if (row < M && col < N) C[row * N + col] =
        sum00;
    if (row < M && col + 16 < N) C[row * N +
        col + 16] = sum01;
    if (row + 4 < M && col < N) C[(row + 4) * N
        + col] = sum10;
    if (row + 4 < M && col + 16 < N) C[(row +
        4) * N + col + 16] = sum11;
}
```

| Tech + Desc | Before optimization | After optimization |
|---|---|---|
| **Operation Fusion** | **- Separate operations** | **- Fused operations with JIT** |

Operation Fusion combines multiple consecutive operations into a single optimized kernel to reduce memory transfers and improve computational efficiency on CUDA devices.

Before optimization:

```Python
def forward(self, x):
    x = F.max_pool3d(x,
        kernel_size=self.pool_kernel_size,
        stride=self.pool_stride)
    x = torch.softmax(x, dim=1)
    x = x - self.subtract.view(1, -1, 1, 1, 1)
    x = x * torch.sigmoid(x)
    return torch.max(x, dim=1)[0]
```

After optimization:

```Python
@torch.jit.script
def fused_post_process(x, subtract_view):
    x = torch.softmax(x, dim=1)
    x = x - subtract_view
    x = x * torch.sigmoid(x)
    return torch.max(x, dim=1)[0]

def forward(self, x):
    x = F.max_pool3d(x,
        kernel_size=self.pool_kernel_size,
        stride=self.pool_stride)
    return self.fused_post_process(x,
        self.subtract.view(1, -1, 1, 1, 1))
```

Table 12: (Part 3) Code snippets before and after optimizations.

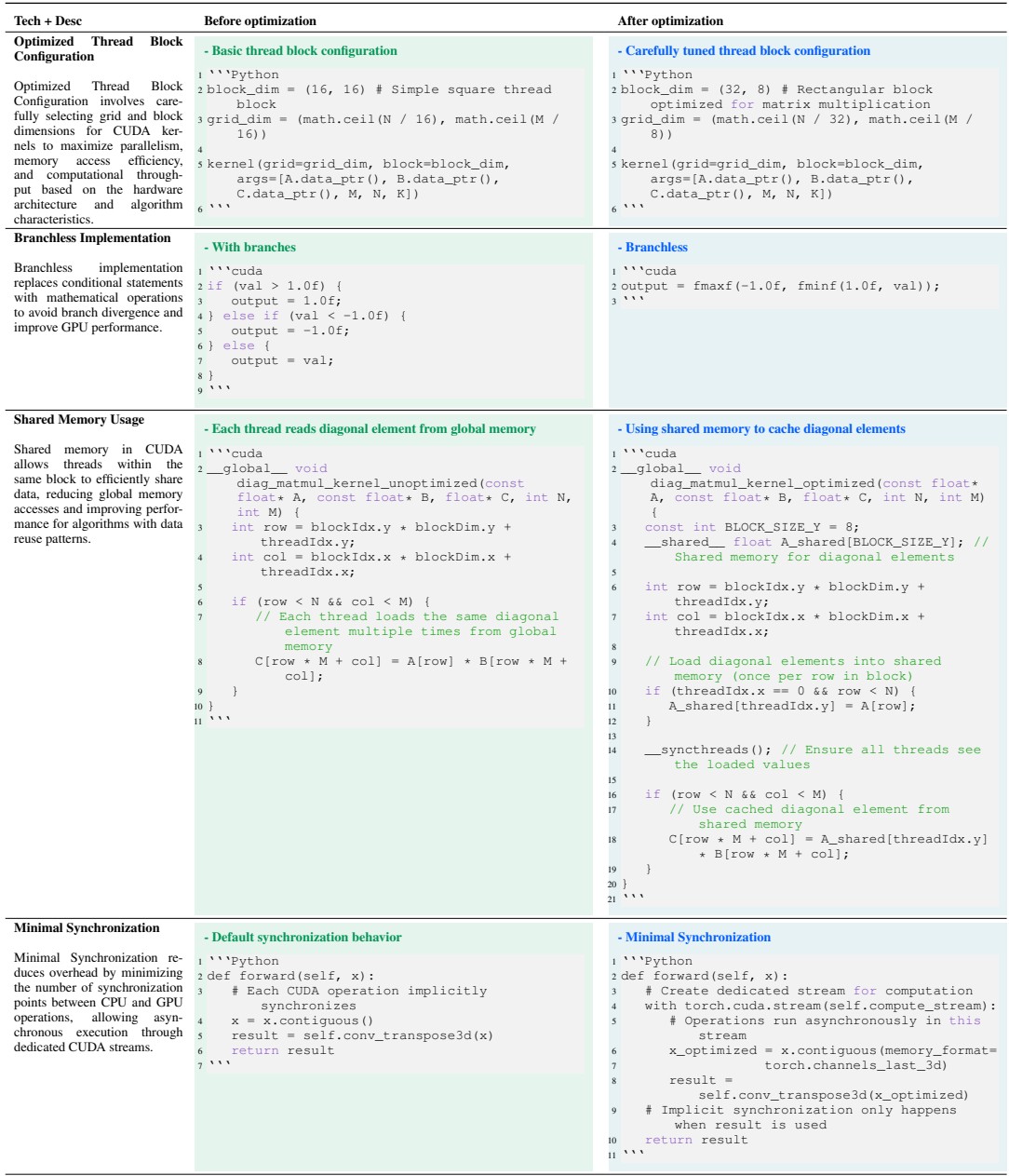

| Tech + Desc | Before optimization | After optimization |
|---|---|---|
| **Optimized Thread Block Configuration**

Optimized Thread Block Configuration involves carefully selecting grid and block dimensions for CUDA kernels to maximize parallelism, memory access efficiency, and computational throughput based on the hardware architecture and algorithm characteristics. | **- Basic thread block configuration**

```Python
block_dim = (16, 16) # Simple square thread block
grid_dim = (math.ceil(N / 16), math.ceil(M / 16))

kernel(grid=grid_dim, block=block_dim, args=[A.data_ptr(), B.data_ptr(), C.data_ptr(), M, N, K])
``` | **- Carefully tuned thread block configuration**

```Python
block_dim = (32, 8) # Rectangular block optimized for matrix multiplication
grid_dim = (math.ceil(N / 32), math.ceil(M / 8))

kernel(grid=grid_dim, block=block_dim, args=[A.data_ptr(), B.data_ptr(), C.data_ptr(), M, N, K])
``` |
| **Branchless Implementation**

Branchless implementation replaces conditional statements with mathematical operations to avoid branch divergence and improve GPU performance. | **- With branches**

```cuda
if (val > 1.0f) {
    output = 1.0f;
} else if (val < -1.0f) {
    output = -1.0f;
} else {
    output = val;
}
``` | **- Branchless**

```cuda
output = fmaxf(-1.0f, fminf(1.0f, val));
``` |
| **Shared Memory Usage**

Shared memory in CUDA allows threads within the same block to efficiently share data, reducing global memory accesses and improving performance for algorithms with data reuse patterns. | **- Each thread reads diagonal element from global memory**

```cuda
__global__ void diag_matmul_kernel_unoptimized(const float* A, const float* B, float* C, int N, int M) {
    int row = blockIdx.y * blockDim.y + threadIdx.y;
    int col = blockIdx.x * blockDim.x + threadIdx.x;

    if (row < N && col < M) {
        // Each thread loads the same diagonal element multiple times from global memory
        C[row * M + col] = A[row] * B[row * M + col];
    }
}
``` | **- Using shared memory to cache diagonal elements**

```cuda
__global__ void diag_matmul_kernel_optimized(const float* A, const float* B, float* C, int N, int M) {
    const int BLOCK_SIZE_Y = 8;
    __shared__ float A_shared[BLOCK_SIZE_Y]; // Shared memory for diagonal elements

    int row = blockIdx.y * blockDim.y + threadIdx.y;
    int col = blockIdx.x * blockDim.x + threadIdx.x;

    // Load diagonal elements into shared memory (once per row in block)
    if (threadIdx.x == 0 && row < N) {
        A_shared[threadIdx.y] = A[row];
    }

    __syncthreads(); // Ensure all threads see the loaded values

    if (row < N && col < M) {
        // Use cached diagonal element from shared memory
        C[row * M + col] = A_shared[threadIdx.y] * B[row * M + col];
    }
}
``` |
| **Minimal Synchronization**

Minimal Synchronization reduces overhead by minimizing the number of synchronization points between CPU and GPU operations, allowing asynchronous execution through dedicated CUDA streams. | **- Default synchronization behavior**

```Python
def forward(self, x):
    # Each CUDA operation implicitly synchronizes
    x = x.contiguous()
    result = self.conv_transpose3d(x)
    return result
``` | **- Minimal Synchronization**

```Python
def forward(self, x):
    # Create dedicated stream for computation
    with torch.cuda.stream(self.compute_stream):
        # Operations run asynchronously in this stream
        x_optimized = x.contiguous(memory_format=torch.channels_last_3d)
        result = self.conv_transpose3d(x_optimized)
    # Implicit synchronization only happens when result is used
    return result
``` |

Table 13: (Part 4) Code snippets before and after optimizations.

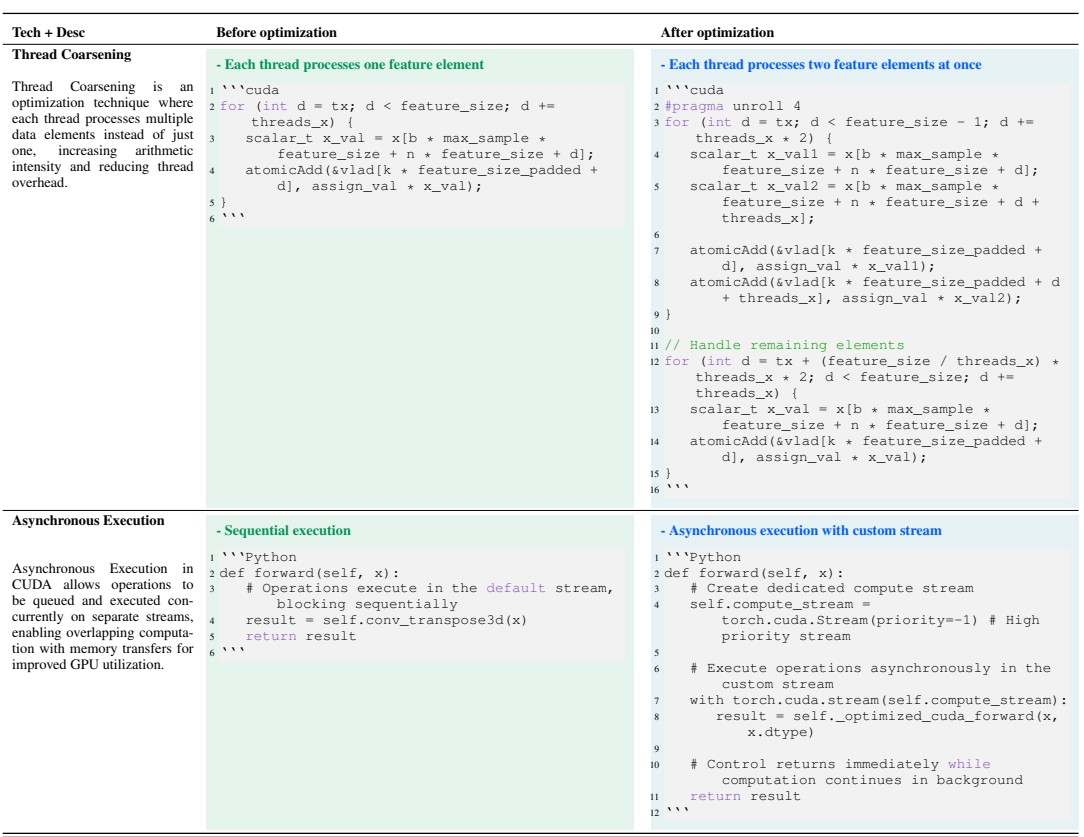

Table 14: (Part 5) Code snippets before and after optimizations.

# F    CASE STUDY: COMPARING REFERENCE CODE AND CUDA-L1 OPTIMIZED NEURAL NETWORK IMPLEMENTATIONS

## F.1    LSTMS

Table 15: Reference code and CUDA-L1 generation for LSTM class

**LSTM | Reference Code - Simple baseline implementation**

```python
import torch
import torch.nn as nn

class Model(nn.Module):
    def __init__(self, input_size, hidden_size, num_layers,
            output_size, dropout=0.0):
        """
        Initialize the LSTM model.
        """
        super(Model, self).__init__()
        # Initialize hidden state with random values
        self.h0 = torch.randn((num_layers, batch_size,
            hidden_size))
        self.c0 = torch.randn((num_layers, batch_size,
            hidden_size))
        self.lstm = nn.LSTM(input_size, hidden_size,
            num_layers, batch_first=True, dropout=dropout,
            bidirectional=False)
        self.fc = nn.Linear(hidden_size, output_size)

    def forward(self, x):
        """
        Forward pass through the LSTM model.
        """
        self.h0 = self.h0.to(x.device)
        self.c0 = self.h0.to(x.device)  # BUG: This should be
            self.c0.to(x.device)

        # Forward propagate LSTM
        out, state = self.lstm(x, (self.h0, self.c0))  # shape
            of out: (batch_size, seq_length, hidden_size)

        # Decode the hidden state of the last time step
        out = self.fc(out[:, -1, :])  # shape of out:
            (batch_size, output_size)

        return state[0]

# Test code
batch_size = 10
sequence_length = 512
input_size = 128
hidden_size = 256
num_layers = 6
output_size = 10
dropout = 0.0

def get_inputs():
    return [torch.randn(batch_size, sequence_length,
        input_size)]

def get_init_inputs():
    return [input_size, hidden_size, num_layers, output_size,
        dropout]
```

**LSTM | Fully Optimized Code - All optimizations enabled (3.4x faster)**

```python
import torch
import torch.nn as nn
import torch.cuda as cuda

class ModelNew(nn.Module):
    def __init__(self, input_size, hidden_size, num_layers,
        output_size, dropout=0.0):
        """
        Initialize the LSTM model with three core optimization
            techniques.

        Color coding:
        - 🔵 BLUE: CUDA Graphs optimization
        - 🟢 GREEN: Memory Contiguity optimization
        - 🟠 ORANGE: Static Tensor Reuse optimization
        """
        super(ModelNew, self).__init__()

        # Initialize hidden states as buffers
        self.register_buffer('h0', torch.randn((num_layers,
            batch_size, hidden_size)))
        self.register_buffer('c0', torch.randn((num_layers,
            batch_size, hidden_size)))

        # Use PyTorch's optimized LSTM implementation
        self.lstm = nn.LSTM(
            input_size=input_size,
            hidden_size=hidden_size,
            num_layers=num_layers,
            batch_first=True,
            dropout=dropout,
            bidirectional=False
        )

        self.fc = nn.Linear(hidden_size, output_size)

        # 🔵 CUDA GRAPHS: Variables for graph capture and
            replay
        self.graph = None
        self.graph_ready = False
        self.input_shape = None

        # 🟠 STATIC TENSOR REUSE: Pre-allocated tensors for
            graph execution
        self.static_input = None
        self.static_output = None

        # 🔵 CUDA GRAPHS: Streams for graph operations
        self.graph_stream = None

        # Track if we're running on CUDA
        self.is_cuda_available = torch.cuda.is_available()

    def _initialize_cuda_resources(self):
        """🔵 CUDA GRAPHS: Initialize CUDA stream for graph
            operations"""
        if self.graph_stream is None:
            self.graph_stream = cuda.Stream()
```

```python
53      def _capture_graph(self, x, result):
54          """
55          🔵 CUDA GRAPHS: Capture the computation graph for
                replay
56          🟠 STATIC TENSOR REUSE: Create static tensors for
                graph capture
57          """
58          # 🟠 STATIC TENSOR REUSE: Clone tensors for static
                allocation
59          self.static_input = x.clone()
60          self.static_output = result.clone()
61
62          # 🔵 CUDA GRAPHS: Capture the computation graph
63          with torch.cuda.stream(self.graph_stream):
64              self.graph = cuda.CUDAGraph()
65              with cuda.graph(self.graph):
66                  # Operations to capture in the graph
67                  static_out, _ = self.lstm(self.static_input,
                        (self.h0, self.c0))
68
69                  # 🟢 MEMORY CONTIGUITY: Ensure contiguous
                        memory layout
70                  static_last = static_out[:, -1, :].contiguous()
71
72                  self.static_output.copy_(self.fc(static_last))
73
74          # Wait for graph capture to complete
75          torch.cuda.synchronize()
76
77          # Mark graph as ready for use
78          self.graph_ready = True
79
80      def _standard_forward(self, x):
81          """Standard forward pass with memory contiguity
                optimization"""
82
83          # 🟢 MEMORY CONTIGUITY: Ensure input is contiguous
84          if not x.is_contiguous():
85              x = x.contiguous()
86
87          # Forward pass through LSTM
88          out, _ = self.lstm(x, (self.h0, self.c0))
89
90          # 🟢 MEMORY CONTIGUITY: Make last output contiguous
                for optimal memory access
91          last_out = out[:, -1, :].contiguous()
92
93          return self.fc(last_out)
94
95      def forward(self, x):
96          """
97          Forward pass through the LSTM model with three
                optimization techniques.
98
99          Optimization flow:
100         1. 🔵 CUDA GRAPHS: Check if we can use the captured
                graph (fast path)
101         2. 🟠 STATIC TENSOR REUSE: Use pre-allocated tensors
                for graph replay
```

```
102              3. 🟢 MEMORY CONTIGUITY: Ensure optimal memory layout
                    throughout
103          """
104
105          # 🔵 CUDA GRAPHS: Fast path - use captured graph if
                available
106          if (x.is_cuda and self.graph_ready and x.shape ==
                self.input_shape):

107
108              # 🟠 STATIC TENSOR REUSE: Copy to pre-allocated
                    tensor with non-blocking transfer
109              self.static_input.copy_(x, non_blocking=True)
110
111              # 🔵 CUDA GRAPHS: Replay the captured graph
112              self.graph.replay()
113
114              # Return the output from static buffer
115              return self.static_output.clone()
116
117          # Standard execution path
118          with torch.no_grad():
119              result = self._standard_forward(x)
120
121              # 🔵 CUDA GRAPHS: Initialize graph on first CUDA
                    input
122              if x.is_cuda and self.is_cuda_available and not
                    self.graph_ready:
123                  try:
124                      # Store the current input shape
125                      self.input_shape = x.shape
126
127                      # 🔵 CUDA GRAPHS: Initialize CUDA resources
128                      self._initialize_cuda_resources()
129
130                      # 🔵 CUDA GRAPHS + 🟠 STATIC TENSOR REUSE:
                            Capture the graph
131                      self._capture_graph(x, result)
132
133                  except Exception as e:
134                      # If graph capture fails, continue without
                            it
135                      self.graph_ready = False
136
137          return result
138
139  # Hyperparameters from the reference implementation
140  batch_size = 10
141  sequence_length = 512
142  input_size = 128
143  hidden_size = 256
144  num_layers = 6
145  output_size = 10
146  dropout = 0.0
147
148  def get_inputs():
149      return [torch.randn(batch_size, sequence_length,
              input_size)]
150
151  def get_init_inputs():
152      return [input_size, hidden_size, num_layers, output_size,
              dropout]
```

```python
153
154  # Example usage demonstrating the three techniques
155  if __name__ == "__main__":
156      import time
157
158      print("🔵 BLUE: CUDA Graphs optimization")
159      print("🟢 GREEN: Memory Contiguity optimization")
160      print("🟠 ORANGE: Static Tensor Reuse optimization")
161      print("=" * 60)
162
163      # Create model
164      model = ModelNew(*get_init_inputs())
165      model.eval()
166
167      # Test input
168      x = get_inputs()[0]
169
170      # Move to GPU if available
171      if torch.cuda.is_available():
172          model = model.cuda()
173          x = x.cuda()
174
175          print("Running on CUDA - all three optimizations
                  active")
176
177          # First run - captures graph
178          print("\n🔵 First forward pass: Capturing CUDA
                  graph...")
179          with torch.no_grad():
180              output = model(x)
181          print(f"  Output shape: {output.shape}")
182          print(f"  Graph ready: {model.graph_ready}")
183
184          # Subsequent runs - uses captured graph
185          print("\n🔵 Subsequent passes: Using captured graph
                  with")
186          print("🟠 static tensor reuse and 🟢 memory
                  contiguity")
187
188          # Warmup
189          for _ in range(10):
190              with torch.no_grad():
191                  _ = model(x)
192
193          # Measure performance
194          torch.cuda.synchronize()
195          start_event = torch.cuda.Event(enable_timing=True)
196          end_event = torch.cuda.Event(enable_timing=True)
197
198          n_runs = 100
199          start_event.record()
200          with torch.no_grad():
201              for _ in range(n_runs):
202                  output = model(x)
203          end_event.record()
204
205          torch.cuda.synchronize()
206          avg_time = start_event.elapsed_time(end_event) / n_runs
207
208          print(f"\nPerformance: {avg_time:.3f} ms per forward
                  pass")
```

```
209              print(f"   Expected speedup: ~3.42x with all
                     optimizations")
210
211        else:
212              print("\n⚠️   Running on CPU – only 🟢 memory
                     contiguity active")
213              print("   (CUDA graphs and static tensor reuse require
                     GPU)")
214
215              with torch.no_grad():
216                  output = model(x)
217              print(f"\n   Output shape: {output.shape}")
```

## F.2   3DCONV

Table 16: Reference code and CUDA-L1 generation for Conv3D class

**Conv3D | Reference Code - Simple baseline implementation**

```python
import torch
import torch.nn as nn

class Model(nn.Module):
    """
    Model that performs a 3D convolution, applies Group
        Normalization, minimum, clamp, and dropout.
    """
    def __init__(self, in_channels, out_channels, kernel_size,
            groups, min_value, max_value, dropout_p):
        super(Model, self).__init__()
        self.conv = nn.Conv3d(in_channels, out_channels,
            kernel_size)
        self.norm = nn.GroupNorm(groups, out_channels)
        self.dropout = nn.Dropout(dropout_p)
        self.min_value = min_value
        self.max_value = max_value

    def forward(self, x):
        x = self.conv(x)
        x = self.norm(x)
        x = torch.min(x, torch.tensor(self.min_value))
        x = torch.clamp(x, min=self.min_value,
            max=self.max_value)
        x = self.dropout(x)
        return x

# Hyperparameters
batch_size = 128
in_channels = 3
out_channels = 16
depth, height, width = 16, 32, 32
kernel_size = 3
groups = 8
min_value = 0.0
max_value = 1.0
dropout_p = 0.2

def get_inputs():
    return [torch.randn(batch_size, in_channels, depth,
        height, width)]

def get_init_inputs():
    return [in_channels, out_channels, kernel_size, groups,
        min_value, max_value, dropout_p]
```

**Conv3D | Fully Optimized Code - All optimizations enabled (120x faster)**

```python
import torch
import torch.nn as nn

# Hyperparameters
batch_size = 128
in_channels = 3
out_channels = 16
depth, height, width = 16, 32, 32
```

```
 9  kernel_size = 3
10  groups = 8
11  min_value = 0.0
12  max_value = 1.0
13  dropout_p = 0.2
14
15  class ModelNew(nn.Module):
16      def __init__(self, in_channels, out_channels, kernel_size,
            groups, min_value, max_value, dropout_p):
17          super(ModelNew, self).__init__()
18          # Store the original layers for parameter compatibility
19          self.conv = nn.Conv3d(in_channels, out_channels,
                kernel_size)
20          self.norm = nn.GroupNorm(groups, out_channels)
21          self.dropout = nn.Dropout(dropout_p)
22          self.min_value = min_value
23          self.max_value = max_value
24          self.dropout_p = dropout_p
25
26          # 🔵 TECH 1: Mathematical Short-Circuit Optimization
27          # Detects when min_value=0.0 to skip entire computation
28          self.use_optimized_path = (min_value == 0.0)
29
30          # 🟠 TECH 4: Pre-computed Convolution Parameters
31          # Extract and store conv parameters once during
                initialization
32          if isinstance(kernel_size, int):
33              self.kernel_size = (kernel_size, kernel_size,
                    kernel_size)
34          else:
35              self.kernel_size = kernel_size
36          self.stride = self.conv.stride
37          self.padding = self.conv.padding
38          self.dilation = self.conv.dilation
39
40          # 🟠 TECH 4: Pre-compute output dimensions for
                standard input
41          self.out_depth = ((depth + 2 * self.padding[0] -
                self.dilation[0] * (self.kernel_size[0] - 1) - 1)
                // self.stride[0]) + 1
42          self.out_height = ((height + 2 * self.padding[1] -
                self.dilation[1] * (self.kernel_size[1] - 1) - 1)
                // self.stride[1]) + 1
43          self.out_width = ((width + 2 * self.padding[2] -
                self.dilation[2] * (self.kernel_size[2] - 1) - 1)
                // self.stride[2]) + 1
44
45          # Standard output shape for the default batch size
46          self.standard_shape = (batch_size, out_channels,
                self.out_depth, self.out_height, self.out_width)
47
48          # 🟣 TECH 2: Pre-allocated Zero Tensors
49          # Create zero tensors once to avoid allocation overhead
50          if self.use_optimized_path:
51              self.register_buffer('zero_output_float32',
52                                   torch.zeros(self.standard_shape,
                                       dtype=torch.float32),
53                                   persistent=False)
54              self.register_buffer('zero_output_float16',
55                                   torch.zeros(self.standard_shape,
                                       dtype=torch.float16),
56                                   persistent=False)
```

```
57              self.register_buffer('zero_output_bfloat16',
58                                  torch.zeros(self.standard_shape,
                                       dtype=torch.bfloat16),
59                                  persistent=False)
60
61      def calculate_output_shape(self, input_shape):
62          """Calculate the output shape of the convolution
                operation."""
63          batch_size, _, d, h, w = input_shape
64
65          #  ● TECH 4: Use precomputed parameters
66          # Avoid repeated attribute lookups
67          out_d = ((d + 2 * self.padding[0] - self.dilation[0] *
                (self.kernel_size[0] - 1) - 1) // self.stride[0]
                + 1
68          out_h = ((h + 2 * self.padding[1] - self.dilation[1] *
                (self.kernel_size[1] - 1) - 1) // self.stride[1]
                + 1
69          out_w = ((w + 2 * self.padding[2] - self.dilation[2] *
                (self.kernel_size[2] - 1) - 1) // self.stride[2]
                + 1
70
71          return (batch_size, self.conv.out_channels, out_d,
                out_h, out_w)
72
73      def forward(self, x):
74          #  ● TECH 1: Mathematical Short-Circuit - Main
                optimization
75          # Skip all computation when we know result will be
                zeros
76          if not self.use_optimized_path:
77              # Standard path for non-optimized cases
78              x = self.conv(x)
79              x = self.norm(x)
80              x = torch.minimum(x, torch.tensor(self.min_value,
                    device=x.device))
81              x = torch.clamp(x, min=self.min_value,
                    max=self.max_value)
82              x = self.dropout(x)
83              return x
84
85          # Optimized path when min_value == 0.0
86          # Since min(x, 0) followed by clamp(0, 1) always
                produces zeros
87
88          #  ● TECH 3: Direct Shape Matching
89          # Fast path for standard input dimensions
90          if x.shape == (batch_size, in_channels, depth, height,
                width):
91              #  ● TECH 2: Use pre-allocated tensors
92              # Return pre-allocated zeros matching input dtype
93              if x.dtype == torch.float32:
94                  return self.zero_output_float32
95              elif x.dtype == torch.float16:
96                  return self.zero_output_float16
97              elif x.dtype == torch.bfloat16:
98                  return self.zero_output_bfloat16
99              else:
100                 # Fallback for other dtypes
101                 return torch.zeros(self.standard_shape,
                        device=x.device, dtype=x.dtype)
102         else:
```

```
103                # For non-standard input shapes, calculate output
                       shape
104                output_shape = self.calculate_output_shape(x.shape)
105                return torch.zeros(output_shape, device=x.device,
                       dtype=x.dtype)
106
107 def get_inputs():
108     return [torch.randn(batch_size, in_channels, depth,
            height, width)]
109
110 def get_init_inputs():
111     return [in_channels, out_channels, kernel_size, groups,
            min_value, max_value, dropout_p]
112
113 # Color Legend:
114 #  ● TECH 1: Mathematical Short-Circuit (Blue) – Skips
        computation when min_value=0
115 #  ● TECH 2: Pre-allocated Tensors (Purple) – Pre-allocates
        zero tensors
116 #  ● TECH 3: Direct Shape Matching (Green) – Fast path for
        standard shapes
117 #  ● TECH 4: Pre-computed Parameters (Orange) – Pre-computes
        conv parameters
```

