# OpenReview forum: "CUDA-L1: Improving CUDA Optimization via  Contrastive Reinforcement Learning"
_ICLR.cc/2026/Conference — ICLR 2026 Poster_

### Official Review · Reviewer_hU7V · 2025-10-26

**Soundness:** 2
**Presentation:** 3
**Contribution:** 2
**Rating:** 4
**Confidence:** 4

**Summary:**

This paper proposes CUDA-L1, a comprehensive framework for automating the optimization of CUDA kernels using large language models (LLMs) enhanced through contrastive reinforcement learning. The framework targets the challenge of improving GPU computing efficiency by enabling LLMs to autonomously generate optimized CUDA implementations that are both correct and performant.
CUDA-L1 adopts a three-stage training pipeline—supervised fine-tuning with data augmentation, self-supervised refinement, and contrastive reinforcement learning—to progressively enhance the model’s ability to reason about, generate, and evaluate CUDA code. Through comparative learning, the model integrates execution feedback directly into its reasoning process, allowing it to learn from past examples of successful and unsuccessful optimizations.
The system is evaluated on a large and diverse set of CUDA kernels across multiple GPU architectures. The paper further contributes a set of CUDA Graph implementations for KernelBench, enhancing community baselines.

**Strengths:**

- The extensive evaluation across all 250 KernelBench kernels—covering multiple GPU architectures and varying complexity levels—provides strong empirical validation of the proposed approach.
- The enrichment of the KernelBench dataset with CUDA Graph implementations represents a valuable and reusable contribution to the research community.
- The identification of non-obvious optimization techniques (e.g., mathematical short-circuiting) highlights the potential of reinforcement learning to uncover previously unexplored performance improvements.

**Weaknesses:**

- The paper conflates CUDA kernel optimization with PyTorch-level program optimization. Case studies such as LSTM and 3D convolution reveal that the “optimized” code primarily alters high-level operator invocation and execution patterns, leaving the underlying CUDA kernels unchanged and thus creating a misleading impression of kernel-level innovation.
- While the paper demonstrates impressive success and speedup rates, it lacks an in-depth analysis of cases where the optimization failed or underperformed.
- The work does not discuss the computational resources required to train the CUDA-L1 system. Given the multi-stage training and RL fine-tuning involved, such costs could be substantial.
- The paper focuses heavily on empirical results but provides limited theoretical explanation for why the contrastive-RL approach outperforms alternatives.
- Although the paper claims that the reward checking model “successfully identifies reward hacking over 60% of the time,” it does not quantify how much this detection reduced actual reward hacking, nor whether the system can recognize previously unseen hacking strategies.
- While the internal baselines and ablations are comprehensive, the work lacks direct comparison with external, published state-of-the-art methods.

**Questions:**

- Given that the paper's core contribution is framed as "CUDA optimization" but the evidence suggests most improvements come from higher-level framework patterns, how do you distinguish between these two distinct types of optimization in your evaluation methodology?
- Since CUDA evolves over time, how would CUDA-L1 handle updates to the CUDA toolkit? Would it require full retraining, or does the model have mechanisms to adapt to new features incrementally?
- The paper builds upon deepseek-v3-671B. How critical is this specific large-scale model to CUDA-L1’s performance? Have experiments been conducted with smaller open-source models, and if so, how does performance degrade?
- The case studies show the model discovering both high-level algorithmic improvements and low-level CUDA implementation optimizations. Does CUDA-L1 include an explicit mechanism to decide the level of abstraction for optimization, or does this behavior emerge naturally from the RL process?
- To aid reader understanding, could the authors include a concrete example (e.g., a filled-in mid-training prompt) in the appendix? While Table 9 outlines the general format, an actual example with real—possibly truncated—code snippets and corresponding performance scores would better illustrate how comparative exemplar guidance works in practice.
- Will the training code, reward-checking implementation, or exemplar-guided prompts be released to facilitate replication by other groups?

---

> ### Author Response · Authors · 2025-12-04
>
> re:Concern that the paper conflates CUDA kernel optimization with PyTorch-level program optimization
>
> Many of the kernels are indeed optimized at the CUDA level. We use PyTorch’s load_inline mechanism, which allows embedding raw CUDA code directly within the Python environment. This design choice is required for compatibility with KernelBench.
>
> While LSTM and 3D convolution are examples where the RL agent chose not to modify CUDA code, this is not representative of the broader results. For most tasks, CUDA kernels are directly rewritten. Below is a minimal example (task: Level 1, Task 5) demonstrating the use of load_inline to incorporate raw CUDA code:
>
> // Python binding
> torch::Tensor matrix_scalar_mul(torch::Tensor input, float scalar) {
>     if (!input.is_cuda()) {
>         input = input.cuda();
>     }
>     return matrix_scalar_mul_cuda(input, scalar);
> }
>
> PYBIND11_MODULE(TORCH_EXTENSION_NAME, m) {
>     m.def("matrix_scalar_mul", &matrix_scalar_mul, "Matrix scalar multiplication");
> }
>
> // Compiling the extension
> try {
>     matrix_scalar_mul_ext = load_inline(
>         name='matrix_scalar_mul_ext',
>         cpp_sources='',
>         cuda_sources=cuda_source,
>         functions=['matrix_scalar_mul'],
>         verbose=False,
>         with_cuda=True
>     );
> }
>
> re: Comparison with Other Baselines
>
> KernelBench contains 250 kernels, and running all of them requires a lot of compute. To our knowledge, no other work reports full results on this benchmark except Sakana AI. However, Sakana’s evaluations were performed on H100, whereas our experiments were conducted on A100, making the results not directly comparable.

---

### Official Review · Reviewer_1yB6 · 2025-10-30

**Soundness:** 3
**Presentation:** 4
**Contribution:** 3
**Rating:** 8
**Confidence:** 3

**Summary:**

This paper introduces CUDA-L1, a multi-stage training pipeline designed to enhance large language models’ (LLMs) capability to generate efficient CUDA kernels. The proposed framework comprises three stages:
- Supervised Fine-Tuning (SFT): The model is trained on synthetic data generated by various state-of-the-art LLMs to increase its exposure to diverse CUDA programming patterns.
- Self-Supervised Learning: The model is further refined by leveraging its own successful generations as additional training data, thereby improving the overall success rate.
 - Contrastive Reinforcement Learning (RL): The model is trained with contrastive rewards applied both at the parameter level and within the LLM context, targeting execution efficiency.
The first two stages primarily aim to maximize correctness and success rate, while the final stage focuses on improving kernel execution speed.

Empirical evaluations demonstrate that CUDA-L1 achieves significant acceleration and near-optimal success rates compared to both human-designed and LLM-based compiler baselines. Moreover, the method shows strong generalizability across different GPU architectures.

**Strengths:**

- The paper is well-written and easy to follow, presenting all critical information clearly in the main text.
- The proposed approach, particularly the contrastive RL stage, is well-motivated. Training the LLM to reason about previous code performance is a thoughtful design, and ablation studies demonstrate its effectiveness.
- Outstanding performance: Empirical results show near 3× improvement on KernelBench with a near-perfect success rate, highlighting the method’s substantial practical impact.

- Comprehensive and transparent analysis: The evaluation and supporting analyses are detailed and informative, with some examples:
  - Detailed performance comparison: Main results include detailed distributions across different difficulty levels in KernelBench.
  - Ablation studies: Systematic studies illustrate the contributions of each stage in the training pipeline. Comparisons with straightforward alternatives (e.g., evolutionary methods or simple RL such as Stage 1+2+GRPO in Table 2) confirm the superiority of the proposed pipeline.
  - Cross-hardware evaluation: Experiments on diverse hardware demonstrate consistent success rates, emphasizing the generalizability advantage of LLM-based methods over traditional human-designed compilers.
  - Transparency: The appendix provides a lot of details, e.g. examples of reward hacking, showing the model attempting to manipulate profiler timers via asynchronous streams.
  - Case studies: Illustrative examples in the appendix demonstrate how the model improves CUDA code generation in practice.

**Weaknesses:**

- It is not entirely clear how the model is evaluated. Specifically, does the evaluation involve iterative refinement over multiple reasoning steps, or is it conducted in a one-shot generation manner? Clarifying this would help assess the robustness of the reported results.

- In Table 1, the average acceleration of nearly 3× appears inconsistent with the P50 speedup of only ~1.2× for the CUDA Graph rows. This raises concerns that the reported average may be skewed by a few poor baseline implementations, resulting in an inflated speedup. Additionally, for Level 2 and Level 3 tasks, which involve more complex operations, the P50 and speedup ratios do not seem to approach a neutral baseline. However, I do not have a better idea on how to present the distribution of acceleration to be more convincing.

**Questions:**

- How is the mean acceleration metric in Table 1 computed? Is it calculated as the mean of per-task acceleration ratios, or as the ratio of mean run times between CUDA-L1 and the baseline? The former could potentially exaggerate the metric, as individual acceleration ratios may vary widely (e.g., from 0 to 100).
- When transferring the model to other GPUs, such as H100, is it necessary to re-run the training pipeline, or can the trained model generalize directly?
- Given that the model is primarily trained in the context of A100 GPUs, does it demonstrate the ability to adapt to new hardware functionalities, such as Tensor Memory Access (TMA) units in H100?
- Are there any plans to open-source the model or training pipeline, allowing the community to build upon this work?

---

### Official Review · Reviewer_rehX · 2025-11-01

**Soundness:** 1
**Presentation:** 2
**Contribution:** 2
**Rating:** 0
**Confidence:** 4

**Summary:**

This paper tackles CUDA code optimization by fine-tuning a large language model through a sequence of supervised fine-tuning (SFT), self-supervised learning, and contrastive reinforcement learning. The authors first construct a dataset for SFT by prompting diverse LLMs with reference PyTorch code from the KernelBench dataset to produce optimized code variants of the original dataset. They then SFT the model on this corpus and subsequently apply self-supervised learning; the model generates code and successfully executed code samples are fed back for further training. Finally, they employ contrastive reinforcement learning driven by code performance scores to refine the model’s preferences toward faster implementations. The resulting model achieves a 3.12× speedup over the KernelBench code.

**Strengths:**

The authors proposed a contrastive reinforcement learning strategy that incorporates reward scores into prompts, potentially enhancing the LLM’s ability to generate optimized code on KernelBench tasks.

**Weaknesses:**

- LLM is both trained and evaluated solely on the same reference code from KernelBench without splitting the training/test dataset. We cannot verify that the finetuned model can create faster optimized versions of code for other tasks or datasets.

- The comparison between CUDA-L1 and the baseline is unfair. CUDA-L1 is fine-tuned on the code that is known to be successfully compiled and executable, benefiting from multiple inference rounds, whereas other baseline LLMs are limited to only five attempts. Since CUDA-L1 requires additional training costs, a proper assessment of the training method should compare it against alternative fine-tuning strategies, not just untuned baselines.

- There is no direct evaluation of the generated CUDA kernels; only the PyTorch code from the KernelBench is measured. Although the end-to-end evaluation may implicitly include the performance of the CUDA kernels, it is not possible to isolate and assess their contribution. The performance improvements observed could instead originate from optimizations in the surrounding Python code (e.g., removing Python overhead). Consequently, the experiment does not provide evidence that the fine-tuned model can optimize CUDA kernels themselves.

- The only novel element in the finetuning pipeline is prompting with a reward score. The methodology does not encode any prior knowledge of CUDA (or PyTorch) into the LLM. Therefore, it seems largely unrelated to CUDA optimization, contrary to its title.

- The authors use terminology that could mislead readers. The second stage of the fine-tuning pipeline should not be described as self-supervised learning but rather as supervised fine-tuning, since failed trials are excluded from the training data. Because this process depends on feedback from an external execution environment, it is more appropriate to regard it as a form of labeling. Moreover, the fine-tuned LLM does not generate optimized CUDA kernels; it produces optimized PyTorch code written in Python, which is fundamentally different from generating efficient CUDA implementations.

**Questions:**

- Is the reward function defined solely in terms of execution time? Are there any components accounting for memory usage or other resource efficiency metrics?

- Why was the KernelBench dataset chosen for this study? KernelBench is designed to evaluate whether an LLM can generate CUDA code from given PyTorch implementations, not to optimize the PyTorch code itself. Therefore, comparing CUDA-L1’s results with the vanilla KernelBench code and reporting speedups appears to be an inappropriate or uninformative evaluation.

---

> ### Author Response · Authors · 2025-12-04
>
> 1re: Concern about training and evaluating on the same reference code
>
> We apologize for the confusion. Our method belongs to the category of test-time reinforcement learning, where the objective is to optimize the tasks in the test set itself, rather than to generalize to unseen tasks. This setting does not fit the traditional training/test split paradigm.
> A recent high-profile example is
> “Olympiad-level formal mathematical reasoning with reinforcement learning” (Nature, 2025),
> where RL is applied directly to solve the given test problems.
> To make this more intuitive: test-time RL is analogous to solving IMO problems directly, and the goal is to solve the given set of tasks as efficiently as possible, rather than learning a model that generalizes to other exams. Similarly, in our CUDA case, if a kernel is widely used, we are primarily interested in optimizing that particular kernel.

---

### Official Review · Reviewer_paTv · 2025-11-01

**Soundness:** 2
**Presentation:** 2
**Contribution:** 2
**Rating:** 2
**Confidence:** 3

**Summary:**

The paper introduces CUDA L1, a three stage system (SFT followed by self supervised filtering followed by contrastive RL) to optimize CUDA code, where the RL stage injects scored prior variants into the prompt to force comparative reasoning and then updates the policy with a GRPO style objective. On KernelBench (250 tasks), the authors report 3.12× mean speedup over the original Pytorch implementation and consistent improvements over torch.compile variants.

**Strengths:**

•	Strong speedups on a community benchmark and across GPUs.

•	Clear prompt/exemplar design and informative case studies.

**Weaknesses:**

•	One of my concerns with the paper is that it does not guarantee the correctness of optimized code. Optimizing code to generate highly optimized CUDA code is well studied in the compiler literature, and many techniques that guarantee the generation of highly optimized, correct code exist. The fact that this proposed method does not guarantee the correctness of the generated code is an important limitation.

•	Automatic code optimization is well studied in compilers. While the comparison with Pytorch torch.compiler is interesting, the paper should also discuss and compare to other state-of-the-art compilers such as TVM, AutoTVM, Ansor, and Triton, which are the de facto baselines for automated GPU optimization.

•	Several wins stem from graph level optimizations (CUDA Graphs, static reuse, stream control) rather than low level kernel transformations. This is fine in practice but should be made explicit. Consider reporting the proportion of gains due to kernel level vs. graph level vs. PyTorch level changes.


•	Contrastive prompts with scored exemplars are compelling, but the policy optimization is standard GRPO. Please clarify more what is algorithmically new.

•	The authors compare to the Pytorch implementations of KernelBench, but it is unclear what backend library is used to accelerate the Pytorch operators. Are the operators accelerated on GPU ? Which library is used to accelerate the operators? Specifying this is very important since this constitutes the baseline and the difference in performance between the different variants is significant.


•	The paper lacks critical training and compute details (GPU hours, steps, batch sizes). Without these, reproducing the reported speedup is unlikely.

**Questions:**

•	What are the full training hyperparameters and compute for each stage?

•	What fraction of improvements stems from kernel level vs graph level vs framework level optimizations?

•	Can you include TVM/Ansor/Triton comparisons?

---

> ### Author Response · Authors · 2025-12-04
>
> re: Guaranteeing correctness of optimized code
>
> We apologize for the confusion. In fact, guaranteeing 100% correctness of optimized GPU kernels is theoretically impossible, because exact functional equivalence cannot always be ensured due to the non-associativity of floating-point arithmetic, where (a+b)+c \neq a+(b+c)
>
> The commonly adopted practice, which is used in both our work and KernelBench (Ouyang et al., 2025), is to validate correctness via allclose, which checks that the numerical differences between the optimized kernel and a trusted reference kernel fall within acceptable floating-point tolerances.
>
> re: Comparison with state-of-the-art compilers (TVM, AutoTVM, Ansor, Triton)
>
> Our work focuses specifically on the CUDA kernels included in the KernelBench dataset, which is the most comprehensive benchmark suite currently available. KernelBench is designed to run entirely within a Python execution environment, which restricts the applicability of certain compiler frameworks. For this reason, we did not include comparisons with TVM, AutoTVM, Ansor, or Triton.
>
> re: Some performance gains arise from graph-level optimizations
>
> We appreciate this observation. We were fully aware of this possibility. indeed, this motivated us to extend the original KernelBench dataset with its graph-level variant, and to report performance improvements separately from graph-level optimizations. This expanded dataset and analysis are described in line 310, page 6.

---

### Meta-Review · Area_Chair_nTnu · 2026-01-09

**Summary:**

This paper proposes CUDA-L1, an automated reinforcement learning (RL) framework for CUDA optimization via a three-stage pipeline (SFT, rejection sampling, test-time RL). The resulting model achieves a 3.12× speedup on KernelBench.

The reviewers have very mixed initial opinions for this submission. Reviewer 1yB6 is in favor of accepting this paper given the novelty in the contrastive RL stage, the “outstanding performance” on KernelBench, as well as “comprehensive and transparent analysis” and "systematic" ablation studies. In contrast, other reviewers raised a few major concerns:

* W1: Lack of comparison with compilers (paTv) or existing approaches on KernelBench (hU7V), which was clarified in the rebuttal.

* W2: Optimized code lacks correctness guarantee (paTv). This was resolved by the additional clarification from the authors on the testing setup (using `.allclose`).

* W3: Mix-up graph level or python code-level optimizations with kernel transformations (paTv, rehX, hU7V), while R-paTv also noted that “this is fine in practice but should be made explicit”, and the authors also provided additional experiments and report performance improvements separately from graph-level optimizations.

* W4: Lack of clear train/test setup (rehX). The authors clarified that their proposed approach is test-time RL.

Given that the authors have adequately resolved most of the reviewers' concerns, I’d recommend acceptance for this submission.

**Reviewer Concerns:**

See above on how the authors responded to the major concerns raised by reviewers.

**Reviewer Scores:**

* R-paTv and R-rehX would have raised their scores since all their major concerns are addressed.

* R-hU7V may likely raise their score given that their concerns are partially addressed. There are a few outstanding items, such as lack of intrinsic evaluation of the reward model or lack of analysis on optimization failure. However, it seems that these concerns are not on the critical path.

---

### Decision · Program_Chairs · 2026-01-26

Accept (Poster)